# ACCELERATING FEDERATED LEARNING WITH QUICK DISTRIBUTED MEAN ESTIMATION

## ABSTRACT

Distributed Mean Estimation (DME), in which $n$ clients communicate vectors to a parameter server that estimates their average, is a fundamental building block in communication-efficient federated learning. In this paper, we improve on previous DME techniques that achieve the optimal $O(1/n)$ Normalized Mean Squared Error (NMSE) guarantee by asymptotically improving the complexity for either encoding or decoding (or both). To achieve this, we formalize the problem in a novel way that allows us to use off-the-shelf mathematical solvers to design the quantization.

## 1 INTRODUCTION

Federated learning McMahan et al. (2017); Kairouz et al. (2019), is a technique to train models across multiple clients without having to share their data. During each training round, the participating clients send their model updates (hereafter referred to as gradients) to a parameter server that calculates their mean and updates the model for the next round. Collecting the gradients from the participating clients is often communication-intensive, which implies that the network becomes a bottleneck. A promising approach for alleviating this bottleneck and thus accelerating federated learning applications is compression . We identify the *Distributed Mean Estimation (DME)* problem as a fundamental building block that is used for that purpose either to directly communicate the gradients Suresh et al. (2017); Konečný & Richtárik (2018); Vargaftik et al. (2021; 2022); Davies et al. (2021) or as part of more complex acceleration mechanisms Richtárik et al. (2021; 2022); Gorbunov et al. (2021); Szlendak et al. (2022); Condat et al. (2022b); Basu et al. (2019); Condat et al. (2022a); Condat & Richtárik (2022); Horváth et al. (2023); Tyurin & Richtárik (2023); He et al. (2023).

DME is defined as follows. Consider $n$ clients with $d$-dimensional vectors (e.g., gradients) to report; each client sends an approximation of its vector to a parameter server (hereafter referred to as 'server') which estimates the vectors' mean. We briefly survey the most relevant and recent related works for DME. Common to these techniques is that they preprocess the input vectors into a different representation that allows for better compression, generally through quantization of the coordinates.

For example, in Suresh et al. (2017), each client, in $O(d \cdot \log d)$ time, uses a Randomized Hadamard Transform (RHT) to preprocess its vector and then applies stochastic quantization. The transformed vector has a smaller coordinate range (in expectation), which reduces the quantization error. The server then aggregates the transformed vectors before applying the inverse transform to estimate the mean, for a total of $O(n \cdot d + d \cdot \log d)$ time. Such a method has a Normalized Mean Squared Error (*NMSE*) that is bounded by $O(\log d/n)$ using $O(1)$ bits per coordinate. Hereafter, we refer to this method as 'Hadamard'. This work also suggests an alternative method that uses entropy encoding to achieve an NMSE of $O(1/n)$, which is optimal. However, entropy encoding is a compute-intensive process that does not efficiently translate to GPU execution, resulting in a slow decode time.

A different approach to DME computes the Kashin's representation Kashin (1977); Lyubarskii & Vershynin (2010) of a client's vector $\overline{x}$ before applying quantization Caldas et al. (2018); Safaryan et al. (2020). Intuitively, this replaces the $d$-dimensional input vector by $O(d)$ coefficients, each bounded by $O(\|\overline{x}\|_2/\sqrt{d})$. Applying quantization to the coefficients instead of the original vectors allows the server to estimate the mean using $O(1)$ bits per coordinate with an $O(1/n)$ *NMSE*. However, computing the coefficients requires applying multiple RHTs, asymptotically slowing down its *encoding* time from Hadamard's $O(d \cdot \log d)$ to $O(d \cdot \log d \cdot \log(n \cdot d))$.

The works of Vargaftik et al. (2021; 2022) transform the input vectors in the same manner as Suresh et al. (2017), but with two differences: (1) clients must use independent transforms; (2) clients use deterministic (*biased*) quantization, derived using existing information-theoretic tools like the Lloyd-Max quantizer, on their transformed vectors. Interestingly, the server still achieves an *unbiased* estimate of each client's input vector after multiplying the estimated vector by a real-valued 'scale'

| Algorithm | Enc. complexity | Dec. complexity | NMSE |
|---|---|---|---|
| **QSGD** Alistarh et al. (2017) | $O(d)$ | $O(n \cdot d)$ | $O(d/n)$ |
| **Hadamard** Suresh et al. (2017) | $O(d \cdot \log d)$ | $O(n \cdot d + d \cdot \log d)$ | $O(\log d/n)$ |
| **Kashin** Caldas et al. (2018); Safaryan et al. (2020) | $O(d \cdot \log d \cdot \log(n \cdot d))$ | $O(n \cdot d + d \cdot \log d)$ | $O(1/n)$ |
| **EDEN-RHT** Vargaftik et al. (2022) | $O(d \cdot \log d)$ | $O(n \cdot d \cdot \log d)$ | $O(1)$ |
| **EDEN-URR** Vargaftik et al. (2022) | $O(d^3)$ | $O(n \cdot d^3)$ | $O(1/n)$ |
| **QUIC-FL (New)** | $O(d \cdot \log d)$ | $O(n \cdot d + d \cdot \log d)$ | $O(1/n)$ |

Table 1: DME worst-case guarantees (without variable-length encoding; see App. B) for $b = O(1)$.

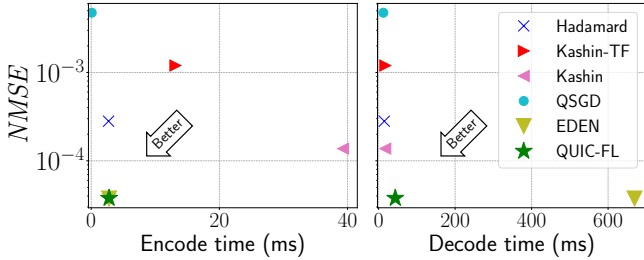

Figure 1: Normalized Mean Squared Error vs. processing time.

(that is sent by the client) and applying the inverse transform. Using uniform random rotations, which RHT approximates, such a process achieves $O(1/n)$ *NMSE* and is empirically more accurate than Kashin's representation. With RHT, their encoding complexity is $O(d \cdot \log d)$, matching that of Suresh et al. (2017). However, since the clients transform their vectors independently of each other (and thus the server must invert their transforms individually, i.e., perform $n$ inverse transforms), the decode time is asymptotically increased to $O(n \cdot d \cdot \log d)$ compared to Hadamard's $O(n \cdot d + d \cdot \log d)$. Further, with RHT the algorithm is biased, and thus its worse-case *NMSE* does not decrease in $1/n$; empirically, it works well for gradient distributions, but we show in Appendix A, there are adversarial cases.

While the above methods suggest aggregating the gradients directly using DME, recent works leverage it as a building block. For example, in EF21 Richtárik et al. (2021), each client sends the compressed difference between its local gradient and local state, and the server estimates the mean to update the global state. Similarly, DIANA Mishchenko et al. (2019) uses DME to estimate the average gradient difference. Thus, better DME techniques can improve their performance (see Appendix J.2). We defer further discussion of frameworks that use DME as a building block to Appendix B.

In this work, we present **Q**uick **U**nb**i**ased **C**ompression for **F**ederated **L**earning (QUIC-FL), a DME method with $O(d \cdot \log d)$ encode and $O(n \cdot d + d \cdot \log d)$ decode times, and the optimal $O(1/n)$ *NMSE*. As summarized in Table 1, QUIC-FL asymptotically improves over the best encoding and/or decoding times of techniques with this *NMSE* guarantee.

In QUIC-FL, each client applies RHT and quantizes its transformed vector using an *unbiased* method we develop to minimize the quantization error. Critically, all clients use *the same* transform, thus allowing the server to aggregate the results before applying a single inverse transform. QUIC-FL's quantization features two new techniques; first, we present Bounded Support Quantization (BSQ), where clients send a small fraction of their largest (transformed) coordinates exactly, thus minimizing the difference between the largest quantized coordinate and the smallest one and thereby the quantization error. Second, we design a near-optimal distribution-aware unbiased quantization. To the best of our knowledge, such a method is not known in the information-theory literature and may be of independent interest.

We implement QUIC-FL in PyTorch Paszke et al. (2019) and TensorFlow Abadi et al. (2015) and evaluate it on different FL tasks (Section 4). We show that QUIC-FL can compress vectors with over 33 million coordinates within 44 milliseconds and is markedly more accurate than existing $O(n \cdot d)$ and $O(n \cdot d + d \cdot \log d)$ decode time approaches such as QSGD Alistarh et al. (2017), Hadamard Suresh et al. (2017), and Kashin Caldas et al. (2018); Safaryan et al. (2020). Compared with DRIVE Vargaftik et al. (2021) and EDEN Vargaftik et al. (2022), QUIC-FL has a competitive NMSE while asymptotically improving the estimation time, as shown in Figure 1. Recent academic and industry sources (e.g., McMahan et al. (2022); Bonawitz et al. (2019)) discuss FL deployments with thousands to tens of thousands of clients per round; thus, this speedup can lead to large savings in time and/or resources. The figure illustrates the encode and decode times vs. NMSE for $b = 4$ bits per coordinate, $d = 2^{20}$ dimensions, and $n = 256$ clients. Our code will be released upon publication.

## 2 PRELIMINARIES

**Notation.** Capital letters denote random variables (e.g., $I_c$) or functions (e.g., $T(\cdot)$); overlines denote vectors (e.g., $\overline{x}_c$); calligraphic letters stand for sets (e.g., $\mathcal{X}_b$) with the exception of $\mathcal{N}$ and $\mathcal{U}$ that denote the normal and uniform distributions; and hats denote estimators (e.g., $\widehat{\overline{x}}_{avg}$).

**Problems and Metrics.** Given a nonzero vector $\overline{x} \in \mathbb{R}^d$, a vector compression protocol consists of a client that sends a message to a server that uses it to estimate $\widehat{\overline{x}} \in \mathbb{R}^d$. The *vector Normalized Mean Squared Error* (*vNMSE*) of the protocol is defined as $\frac{\mathbb{E}\left[\left\|\widehat{\overline{x}} - \overline{x}\right\|_2^2\right]}{\|\overline{x}\|_2^2}$ .

The above generalizes to Distributed Mean Estimation (DME), where each of $n$ clients has a nonzero vector $\overline{x}_c \in \mathbb{R}^d$, where $c \in \{0, \dots, n-1\}$, that they compress and communicate to a server. We are interested in minimizing the *Normalized Mean Squared Error* (*NMSE*), defined as $\frac{\mathbb{E}\left[\left\|\widehat{\overline{x}}_{avg} - \frac{1}{n}\sum_{c=0}^{n-1} \overline{x}_c\right\|_2^2\right]}{\frac{1}{n} \cdot \sum_{c=0}^{n-1} \|\overline{x}_c\|_2^2}$ , where $\widehat{\overline{x}}_{avg}$ is our estimate of the average $\frac{1}{n} \cdot \sum_{c=0}^{n-1} \overline{x}_c$. For unbiased algorithms and independent estimates, we that $NMSE = vNMSE/n$.

**Randomness.** We use *global* (common to all clients and the server) and *client-specific* shared randomness (one client and server). Client-only randomness is called *private*.

## 3 THE QUIC-FL ALGORITHM

We first describe our design goals in Section 3.1. Then, in Sections 3.2 and 3.3, we successively present two new tools we have developed to achieve our goals, namely, *bounded support quantization* and *distribution-aware unbiased quantization*. In Section 3.4, we present QUIC-FL's pseudocode and discuss its properties and guarantees. Finally, in Section 3.5, we overview additional optimizations.

### 3.1 DESIGN GOALS

We aim to develop a DME technique that requires less computational overhead while achieving the same accuracy at the same compression level as the best previous techniques.

As shown by recent works Suresh et al. (2017); Lyubarskii & Vershynin (2010); Caldas et al. (2018); Safaryan et al. (2020); Vargaftik et al. (2021; 2022), a preprocessing stage that transforms each client's vector to a vector with a different distribution (such as applying a uniform random rotation or RHT) can lead to smaller quantization errors and asymptotically lower $NMSE$. However, in existing DME techniques that achieve the asymptotically optimal $NMSE$ of $O(1/n)$, such preprocessing incurs a high computational overhead on either the clients (i.e., Lyubarskii & Vershynin (2010); Caldas et al. (2018); Safaryan et al. (2020)) or the server (i.e., Lyubarskii & Vershynin (2010); Caldas et al. (2018); Safaryan et al. (2020); Vargaftik et al. (2021; 2022)). The question is then how to preserve the appealing $NMSE$ of $O(1/n)$ but reduce the computational burden?

In QUIC-FL, similarly to previous DME techniques, we use a preprocessing stage[1] where each client applies a uniform random rotation on its input vector. After the rotation, the coordinates' distribution approaches independent normal random variables for high dimensions Vargaftik et al. (2021). We use our knowledge of the resulting distribution to devise a fast and near-optimal unbiased quantization scheme that both preserves the appealing $O(1/n)$ $NMSE$ guarantee and is asymptotically faster than existing DME techniques with similar $NMSE$ guarantees. A particularly important aspect of our scheme is that we can avoid decompressing each client's compressed vector at the server by having all clients use the same rotation (determined by shared randomness), so that the server can directly sum the compressed results and perform a single inverse rotation.

### 3.2 BOUNDED SUPPORT QUANTIZATION

Our first contribution is the introduction of *bounded support quantization* (BSQ). For a parameter $p \in (0, 1]$, we pick a threshold $t_p$ such that up to $d \cdot p$ values can fall outside $[-t_p, t_p]$. BSQ separates the vector into two parts: the small values in the range $[-t_p, t_p]$, and the remaining (large) values. The large values are sent exactly (matching the precision of the input), whereas the small values are stochastically quantized and sent using a small number of bits each. This approach decreases the error of the quantized values by bounding their support at the cost of sending a small number of values exactly.

---

[1]In Section 3.5, move to the computationally efficient RHT instead, while preserving Table 1's guarantees.

For the exactly sent values, we also need to send their indices. There are different ways to do so. For example, it is possible to encode these indices using $\log \binom{d}{d \cdot p} \approx d \cdot p \cdot \log(1/p)$ bits at the cost of higher complexity. When the $d \cdot p$ indices are uniformly distributed (which will be essentially our case later), then delta coding methods can be applied (see, e.g., Section 2.3 of Vaidya et al. (2022)). Alternatively, we can send these indices without any additional encoding using $d \cdot p \cdot \lceil \log d \rceil$ bits (i.e., $\lceil \log d \rceil$ bits per transmitted index) or transmit a bit-vector with an indicator for each value whether it is exact or quantized. Empirically, sending the indices using $\lceil \log d \rceil$ bits each without encoding is most useful, as $p \cdot \log d \ll 1$ in our settings, resulting in fast processing time and small bandwidth overhead.

In Appendix C, we prove that BSQ admits a worst-case *NMSE* of $\frac{1}{n \cdot p \cdot (2^b - 1)^2}$ when using $b$ bits per quantized value. In particular, when $p$ and $b$ are constants, we get an *NMSE* of $O(1/n)$ with encoding and decoding times of $O(d)$ and $O(n \cdot d)$, respectively.

However, the linear dependence on $p$ means that the hidden constant in the $O(1/n)$ *NMSE* is often impractical. For example, if $p = 2^{-5}$ and $b = 1$, we need three bits per value on average: two for sending the exact values and their indices (assuming values are single precision floats and indices are 32-bit integers) and another for stochastically quantizing the remaining values using 1-bit stochastic quantization. In turn, we get an *NMSE* bound of $\frac{1}{n \cdot 2^{-5} \cdot (2^1 - 1)^2} = 32/n$.

In the following, we show that combining BSQ with our chosen random rotation preprocessing allows us to get an $O(1/n)$ *NMSE* with a much lower constant for small values of $p$. For example, a basic version of QUIC-FL with $p = 2^{-9}$ and $b = 1$ can reach an *NMSE* of $8.58/n$, a $3.72\times$ improvement despite using $2.66\times$ less bandwidth (i.e., $1.125$ bits per value instead of $3$).

## 3.3 DISTRIBUTION-AWARE UNBIASED QUANTIZATION

The first step towards our goal involves randomly rotating and scaling an input vector and then using BSQ to send values (rotated and scaled coordinates) outside the range $[-t_p, t_p]$ exactly. The values in the range $[-t_p, t_p]$ are sent using stochastic quantization, which ensures unbiasedness for any choice of quantization-values that cover that range. Now we seek quantization-values that minimize the estimation variance and thereby the *NMSE*. We take advantage of the fact that, after randomly rotating a vector $\overline{x} \in \mathbb{R}^d$ and scaling it by $\sqrt{d}/\|\overline{x}\|_2$, the rotated and scaled coordinates approach the distribution of independent normal random variables $\mathcal{N}(0, 1)$ as $d$ increases Vargaftik et al. (2021; 2022). We thus choose to optimize the quantization-values for the normal distribution and later show that it yields a near-optimal quantization for the actual rotated coordinates (see Appendix D for further discussion). That is, since we know both the distribution of the coordinates after the random rotation and scaling and we know the range of the values we are stochastically quantizing, we can design an unbiased quantization scheme that is optimized for this specific distribution rather than using, e.g., the standard approach of uniformly sized intervals.

Formally, for $b$ bits per quantized value and a BSQ parameter $p$, we find the set of quantization-values $\mathcal{Q}_{b,p}$ that minimizes the estimation variance of the random variable $Z \mid Z \in [-t_p, t_p]$ where $Z \sim \mathcal{N}(0, 1)$, after stochastically quantizing it to a value in $\mathcal{Q}_{b,p}$ (i.e., the quantization is unbiased). Then, we show how to use this precomputed set of quantization-values $\mathcal{Q}_{b,p}$ on any preprocessed vector.

Consider parameters $p$ and $b$ and let $\mathcal{X}_b = \{0, \ldots, 2^b - 1\}$. Then, for a *message* $x \in \mathcal{X}_b$, we denote by $S(z, x)$ the probability that the *sender* quantizes a value $z \in [-t_p, t_p]$ to $R(x)$, the value that the *receiver* associates with $x$. With these notations at hand, we solve the following optimization problem to find the set $\mathcal{Q}_{b,p}$ that minimizes the estimation variance (we are omitting the constant factor $1/\sqrt{2\pi}$ in the normal distribution's pdf from the minimization as it does not affect the solution):

$$\underset{S,R}{\text{minimize}} \int_{-t_p}^{t_p} \sum_{x \in \mathcal{X}_b} S(z, x) \cdot (z - R(x))^2 \cdot e^{\frac{-z^2}{2}} dz \qquad \text{subject to}$$

$$(\textit{Unbiasedness}) \quad \sum_{x \in \mathcal{X}_b} S(z, x) \cdot R(x) = z \quad \forall z \in [-t_p, t_p]$$

$$(\textit{Probability}) \quad \sum_{x \in \mathcal{X}_b} S(z, x) = 1 \quad \forall z \in [-t_p, t_p], \qquad S(z, x) \geq 0 \quad \forall z \in [-t_p, t_p], \, x \in \mathcal{X}_b$$

Observe that $\mathcal{Q}_{b,p} = \{R(x) \mid x \in \mathcal{X}_b\}$ is the set of quantization-values that we are seeking. We note that the problem is known to be non-convex for any $b \geq 2$ (Faghri et al., 2020, Appendix B).

While there exist solutions to this problem *excluding* the unbiasedness constraint (e.g., the Lloyd-Max Scalar Quantizer Lloyd (1982); Max (1960)), we are unaware of existing methods for solving the above problem analytically. Instead, we propose a discrete relaxation, allowing us to approach the problem with a *solver*.[2] To that end, we discretize the problem by approximating the truncated normal distribution using a finite set of $m$ *quantiles*. Denote $\mathcal{I}_m = \{0, \ldots, m-1\}$ and let $Z \sim \mathcal{N}(0, 1)$. Then, $\mathcal{A}_{p,m} = \{A_{p,m}(i) \mid i \in \mathcal{I}_m\}$, where the quantile $\mathcal{A}_{p,m}(i)$ satisfies

$$\Pr\left[Z \leq \mathcal{A}_{p,m}(i) \mid Z \in [-t_p, t_p]\right] = \tfrac{i}{m-1}.$$

We find it convenient to denote $S'(i, x) = S(\mathcal{A}_{p,m}(i), x)$. Accordingly, the discretized unbiased quantization problem is defined as (we omit the $1/m$ constant as it does not affect the solution):

$$\underset{S', R}{\text{minimize}} \sum_{i \in \mathcal{I}_m, x \in \mathcal{X}_b} S'(i, x) \cdot (\mathcal{A}_{p,m}(i) - R(x))^2 \qquad \text{subject to}$$

$$(Unbiasedness) \sum_{x \in \mathcal{X}_b} S'(i, x) \cdot R(x) = \mathcal{A}_{p,m}(i) \quad \forall \, i \in \mathcal{I}_m$$

$$(Probability) \sum_{x \in \mathcal{X}_b} S'(i, x) = 1 \quad \forall \, i \in \mathcal{I}_m, \qquad S'(i, x) \geq 0 \quad \forall \, i \in \mathcal{I}_m, \; x \in \mathcal{X}_b$$

The solution to this optimization problem yields the set of quantization-values $\mathcal{Q}_{b,p} = \{R(x) \mid x \in \mathcal{X}_b\}$ we are seeking. A value $z \in [-t_p, t_p]$ (not just the quantiles) is then stochastically quantized to one of the two nearest values in $\mathcal{Q}_{b,p}$. Such quantization is optimal for a fixed set of quantization-values, so we do not need $S$ at this point.

Unlike in vanilla BSQ (Section 3.2), in QUIC-FL, as implied by the optimization problem, the number of values that fall outside the range $[-t_p, t_p]$ may slightly deviate from $d \cdot p$ (and our guarantees are unaffected by this). This is because we precompute the optimal quantization-values set $\mathcal{Q}_{b,p}$ for a given $b$ and $p$ and set $t_p$ according to the $\mathcal{N}(0, 1)$ distribution. In turn, this allows the clients to use $\mathcal{Q}_{b,p}$ when encoding rather than compute $t_p$ and then $\mathcal{Q}_{b,p}$ for each preprocessed vector separately. This results in a near-optimal quantization for the actual rotated and scaled coordinates, in the sense that: (1) for large $d$ values, the distribution of the rotated and scaled coordinates converges to that of independent normal random variables; (2) for large $m$ values, the discrete problem converges to the continuous one.

### 3.4 Putting it all together

The pseudo-code of QUIC-FL appears in Algorithm 1. As mentioned, we use the uniform random rotation as a preprocessing stage done by the clients. Crucially, similarly to Suresh et al. (2017), and unlike in Vargaftik et al. (2021; 2022), all clients use the same rotation, which is a key ingredient in achieving fast decoding complexity.

To compute this rotation (and its inverse by the server), the parties rely on *global shared randomness* as mentioned in Section 2. In practice, having shared randomness only requires the round's participants and the server to agree on a pseudo-random number generator seed, which is standard practice.

**Clients.** Each client $c$ uses global shared randomness to compute its rotated vector $T(\overline{x}_c)$. Importantly, all clients use the same rotation. As discussed, for large dimensions, the distribution of each entry in the rotated vector converges to $\mathcal{N}(0, \|\overline{x}_c\|_2^2 / d)$. Thus, $c$ normalizes it by $\sqrt{d}/\|\overline{x}_c\|_2$ so the values of $\overline{Z}_c$ are approximately distributed as $\mathcal{N}(0, 1)$ (line 1). (Note that we do *not* assume the values are *actually* normally distributed; this is *not* required for our algorithm or our analysis.) Next, the client divides the preprocessed vector into large and small values (lines 2-4). The small values (i.e., whose absolute value is smaller than $t_p$) are stochastically quantized (i.e., in an unbiased manner) to values in the precomputed set $\mathcal{Q}_{b,p}$. We implement $\mathcal{Q}_{b,p}$ as an array where $\mathcal{Q}_{b,p}[x]$ stands for the $x$'th quantization-value; this allows us to transmit just the quantization-value indices over the network (line 5). Finally, each client sends to the server the vector's norm $\|\overline{x}_c\|_2$, the indices $\overline{X}_c$ of the quantization-values of $\overline{V}_c$ (i.e., the small values), and the exact large values with their indices in $\overline{Z}_c$ (line 6).

**Server.** For each client $c$, the server uses $\overline{X}_c$ to look up the quantization-values $\widehat{\overline{V}}_c$ of the small coordinates (line 8) and constructs the estimated scaled rotated vector $\widehat{\overline{Z}}_c$ using $\widehat{\overline{V}}_c$ and the accurate

---

[2]We use the Gekko Beal et al. (2018) software package that provides a Python wrapper to the APMonitor Hedengren et al. (2014) environment, running the solvers IPOPT IPO and APOPT APO.

---

**Algorithm 1** QUIC-FL

---

**Input:** Bit budget $b$, BSQ parameter $p$, and their threshold $t_p$ and precomputed quantization-values $\mathcal{Q}_{b,p}$.

---

**Client $c$:**

1: $\overline{Z}_c \leftarrow \frac{\sqrt{d}}{\|\overline{x}_c\|_2} \cdot T(\overline{x}_c)$

2: $\overline{U}_c \leftarrow \{\overline{Z}_c[i] \mid |\overline{Z}_c[i]| > t_p\}$

3: $\overline{I}_c \leftarrow \{i \mid |\overline{Z}_c[i]| > t_p\}$

4: $\overline{V}_c \leftarrow \{\overline{Z}_c[i] \mid |\overline{Z}_c[i]| \leq t_p\}$

5: $\overline{X}_c \leftarrow$ Stochastically quantize $\overline{V}_c$ using $\mathcal{Q}_{b,p}$

6: Send $(\|\overline{x}_c\|_2, \overline{U}_c, \overline{I}_c, \overline{X}_c)$ to server

**Server:**

7: For all $c$:

8: $\quad \widehat{\overline{V}}_c \leftarrow \{\mathcal{Q}_{b,p}[x] \text{ for } x \text{ in } \overline{X}_c\}$

9: $\quad \widehat{\overline{Z}}_c \leftarrow$ Merge $\widehat{\overline{V}}_c$ and $(\overline{U}_c, \overline{I}_c)$

10: $\widehat{\overline{Z}}_{avg} \leftarrow \frac{1}{n} \cdot \sum_{c=0}^{n-1} \frac{\|\overline{x}_c\|_2}{\sqrt{d}} \cdot \widehat{\overline{Z}}_c$

11: $\widehat{\overline{x}}_{avg} \leftarrow T^{-1}\left(\widehat{\overline{Z}}_{avg}\right)$

---

information about the large coordinates $\overline{U}_c$ and their indices $\overline{I}_c$ (line 9). Then, the server computes the estimate $\widehat{\overline{Z}}_{avg}$ of the average rotated and scaled vector by averaging the reconstructed clients' scaled and rotated vectors and multiplying the results by the inverse scaling factor $\frac{\|\overline{x}_c\|_2}{\sqrt{d}}$ (line 10). Finally, the server performs *a single* inverse rotation using the global shared randomness to obtain the estimate of the mean vector $\widehat{\overline{x}}_{avg}$ (line 11).

In Appendix E, we formally establish the following error guarantee for QUIC-FL (i.e., Algorithm 1).

**Theorem 3.1.** *Let $Z \sim \mathcal{N}(0,1)$ and let $\widehat{Z}$ be its estimation by our distribution-aware unbiased quantization scheme. Then, for any number of clients $n$ and any set of $d$-dimensional input vectors $\{\overline{x}_c \in \mathbb{R}^d \mid c \in \{0, \ldots, n-1\}\}$, we have that QUIC-FL's NMSE respects*

$$NMSE = \frac{1}{n} \cdot \mathbb{E}\left[\left(Z - \widehat{Z}\right)^2\right] + O\left(\frac{1}{n} \cdot \sqrt{\frac{\log d}{d}}\right).$$

The theorem accounts for the cost of quantizing the actual rotated and scaled coordinates (which are not independent and follow a shifted-beta distribution) instead of independent and truncated normal variables. The difference manifests in the $O(1/n \cdot \sqrt{\log d/d}) = O(1/n)$ term; this quickly decays with the dimension and number of clients.

As the theorem suggests, $NMSE \approx \frac{1}{n} \cdot \mathbb{E}[(Z - \widehat{Z})^2]$ for QUIC-FL in settings of interest. Moreover,

$$\mathbb{E}\left[\left(Z - \widehat{Z}\right)^2\right] = \mathbb{E}\left[\left(Z - \widehat{Z}\right)^2 \mid Z \in [-t_p, t_p]\right] \cdot \Pr[Z \in [-t_p, t_p]]$$
$$+ \mathbb{E}\left[\left(Z - \widehat{Z}\right)^2 \mid Z \notin [-t_p, t_p]\right] \cdot \Pr[Z \notin [-t_p, t_p]] ,$$

where the first summand is exactly the quantization error of our distribution-aware unbiased BSQ, and the second summand is $0$ as such values are sent exactly. This means that for any $b$ and $p$, we can exactly compute $\mathbb{E}[(Z - \widehat{Z})^2]$ given the solver's output (i.e., the precomputed quantization values). For example, it is $\approx 8.58$ for $b = 1$ and $p = 2^{-9}$. Another important corollary of Theorem 3.1 is that the convergence speed with QUIC-FL matches the vanilla SGD since its estimates are unbiased and with an $O(1/n)$ NMSE (e.g., see Remark 5 in Karimireddy et al. (2019)).

### 3.5 OPTIMIZATIONS

We introduce two optimizations for QUIC-FL: we further reduce *NMSE* with client-specific shared randomness and then accelerate the processing time via the randomized Hadamard transform.

**QUIC-FL with client-specific shared randomness.** Past works (e.g., Ben Basat et al. (2021); Chen et al. (2020); Roberts (1962b)) on optimizing the quantization-bandwidth tradeoff show the benefit of using *shared randomness* to reduce the quantization error. Here, we show how to leverage this (client-specific) shared randomness to design near-optimal quantization of the rotated and scaled vector.

To that end, in Appendix F, we first extend our optimization problem to allow client-specific shared randomness and then derive the related discretized problem. Importantly, we also discretize the client-specific shared randomness where each client, for each rotated and quantized coordinate, uses a shared random $\ell$-bit value $H \sim \mathcal{U}[\mathcal{H}_l]$ where $\mathcal{H}_\ell = \{0, \ldots, 2^\ell - 1\}$.

The resulting optimization problem is given as follows (additions are highlighted in red):

$$\underset{S',R}{\text{minimize}} \sum_{h \in \mathcal{H}_\ell,\, i \in \mathcal{I}_m,\, x \in \mathcal{X}_b} S'(h,i,x) \cdot (\mathcal{A}_{p,m}(i) - R(h,x))^2 \qquad \text{subject to}$$

$$(\textit{Unbiasedness}) \; \frac{1}{2^\ell} \cdot \sum_{h \in \mathcal{H}_\ell,\, x \in \mathcal{X}_b} S'(h,i,x) \cdot R(h,x) = \mathcal{A}_{p,m}(i) \qquad\qquad \forall i \in \mathcal{I}_m$$

$$(\textit{Probability}) \sum_{x \in \mathcal{X}_b} S'(h,i,x) = 1 \forall h \in \mathcal{H}_\ell, i \in \mathcal{I}_m, \quad S'(h,i,x) \geq 0 \quad \forall h \in \mathcal{H}_\ell, i \in \mathcal{I}_m,\ x \in \mathcal{X}_b$$

Here $S'(h,i,x) = S(h, \mathcal{A}_{p,m}(i), x)$ represents the probability that the sender sends the message $x \in \mathcal{X}_b$ given the shared randomness value $h$ for the input value $\mathcal{A}_{p,m}(i)$. Similarly, $R(h,x)$ is the value the receiver associates with the message $x$ when the shared randomness is $h$. We explain how to use $R(h,x)$ to determine the appropriate message for the sender on a general input $z$, along with further details, in Appendix F. We note that Theorem 3.1 trivially applies to QUIC-FL with client-specific shared randomness as this only lowers the quantization's expected squared error, i.e., $\mathbb{E}[(Z - \widehat{Z})^2]$, and thus the resulting *NMSE*.

Here, we provide an example based on the solver's solution for the case of using a single shared random bit (i.e., $H \sim \mathcal{U}[\mathcal{H}_1]$), a single-bit message ($b = 1$), and $p = 2^{-9}$ ($t_p \approx 3.097$); We can then use the following algorithm, where $X$ is the sent message and $\alpha = 0.7975, \beta = 5.397$ are constants:

$$X = \begin{cases} 1 & \text{if } H = 0 \text{ and } Z \geq 0 \\ 0 & \text{if } H = 1 \text{ and } Z < 0 \\ Bernoulli(\frac{2Z}{\alpha+\beta}) & \text{If } H = 1 \text{ and } Z \geq 0 \\ 1 - Bernoulli(\frac{-2Z}{\alpha+\beta}) & \text{If } H = 0 \text{ and } Z < 0 \end{cases} \qquad \widehat{Z} = \begin{cases} -\beta & \text{if } H = X = 0 \\ -\alpha & \text{if } H = 1 \text{ and } X = 0 \\ \alpha & \text{If } H = 0 \text{ and } X = 1 \\ \beta & \text{If } H = X = 1 \end{cases}.$$

For example, consider $Z = 1$, and recall that $H = 0$ w.p. $1/2$ and $H = 1$ otherwise. Then:

- If $H = 0$, we have $X = 1$ and thus $\widehat{Z} = \alpha$.
- If $H = 1$, then $X = 1$ w.p. $\frac{2}{\alpha+\beta}$ and we get $\widehat{Z} = \beta$. Otherwise (if $X = 0$), we get $\widehat{Z} = -\alpha$.

Indeed, we have that the estimate is unbiased since:

$$\mathbb{E}[\widehat{Z} \mid Z = 1] = \tfrac{1}{2} \cdot \alpha + \tfrac{1}{2} \cdot \left( \tfrac{2}{\alpha+\beta} \cdot \beta + \tfrac{\alpha+\beta-2}{\alpha+\beta} \cdot (-\alpha) \right) = 1.$$

We next calculate the expected squared error (by symmetry, we integrate over positive $z$):

$$\mathbb{E}\left[(Z - \widehat{Z})^2\right] = \sqrt{\tfrac{2}{\pi}} \left( \int_0^{t_p} \tfrac{1}{2} \cdot \left( (z - \alpha)^2 + \tfrac{2z}{\alpha+\beta} \cdot (z-\beta)^2 + \tfrac{\alpha+\beta-2z}{\alpha+\beta} \cdot (z+\alpha)^2 \right) \cdot e^{-z^2/2} dz \right) \approx 3.29.$$

Observe that it is significantly lower than the 8.58 quantization error obtained without shared randomness. As we illustrate (Figure 2), the error further decreases when using more shared random bits.

**Accelerating QUIC-FL with RHT.** Similarly to previous algorithms that use random rotations as a preprocessing state (e.g., Suresh et al. (2017); Vargaftik et al. (2021; 2022)) we propose to use the Randomized Hadamard Transform (RHT) Ailon & Chazelle (2009) instead of uniform random rotations. Although RHT does not induce a uniform distribution on the sphere, it is considerably more efficient to compute, and, under mild assumptions, the resulting distribution is close to that of a uniform random rotation Vargaftik et al. (2021). Nevertheless, we are interested in establishing how using RHT instead of a uniform random rotation affects the formal guarantees of QUIC-FL.

As shown in Appendix G, QUIC-FL with RHT remains unbiased and has the same asymptotic guarantee as with random rotations, albeit with a larger constant (constant factor increases in the fraction of exactly sent values and *NMSE*). See also Appendix D for further discussion and references.

We note that these guarantees are still stronger than those of DRIVE Vargaftik et al. (2021) and EDEN Vargaftik et al. (2022), which only prove RHT bounds for vectors whose coordinates are sampled i.i.d. from a distribution with finite moments, and are not applicable to adversarial vectors.

For example, when $p = 2^{-9}$ and we use $\ell = 4$ shared random bits per quantized coordinate, our analysis shows that the *NMSE* for $b = 1, 2, 3, 4$ is bounded by $4.831/n, 0.692/n,$ $0.131/n, 0.0272/n$, accordingly, and that the expected number of coordinates outside $[-t_p, t_p]$ is bounded by $3.2 \cdot p \cdot d \approx 0.006 \cdot d$. We note that this result does not have the $O\left(1/n \cdot \sqrt{\log d/d}\right)$

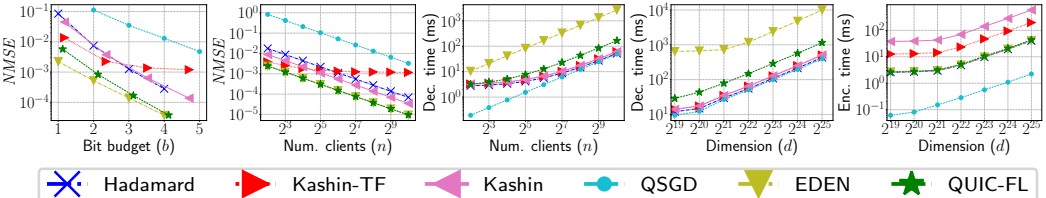

Figure 2: The *NMSE* of QUIC-FL (with $n = 256$ clients) as a function of the bit budget $b$, fraction $p$, and shared random bits $\ell$. In the leftmost figure, $p = 2^{-9}$, while the other two use $b = 4$.

Figure 3: Comparison to alternatives with $n$ clients that have the same $LogNormal(0, 1)$ input vector. The default values are $n = 256$ clients, $b = 4$ bit budget, and $d = 2^{20}$ dimensions.

additive *NMSE* term. The reason is that we directly analyze the error for the Hadamard-rotated coordinates (whereas Theorem 3.1 relies on analyzing the error in quantizing normal variables and factoring in the difference in distributions). In particular, we get that for $p = 2^{-9}, b \in \{1, 2, 3\}$, running QUIC-FL with Hadamard and $(b + 1 + 2.2 \cdot p) \approx b + 1.0043$ bits per coordinate has lower *NMSE* than $b$-bits QUIC-FL with uniform random rotation. That is, one can compensate for the increased error caused by using RHT by adding one bit per coordinate. In practice, as shown in the evaluation, the actual performance is (as one might expect) actually close to the theoretical results for uniform random rotations; improving the bounds is left as future work.

Finally, Table 1 summarizes the theoretical guarantees of QUIC-FL in comparison to state-of-the-art DME techniques. The encoding complexity of QUIC-FL is dominated by RHT and is done in $O(d \cdot \log d)$ time. The decoding of QUIC-FL only requires the addition of all estimated rotated clients' vectors and *a single* inverse RHT transform resulting in $O(n \cdot d + d \cdot \log d)$ time. As mentioned, the *NMSE* with RHT remains $O(1/n)$. Observe that QUIC-FL has an *asymptotic* speed improvement either at the clients or the server among the techniques that achieve $O(1/n)$ *NMSE*.

**A lower bound on the continuous problem.** QUIC-FL obtains a solution for the above problem via the discretization of the distribution and shared randomness. To obtain a lower bound on the *vNMSE* of the continuous problem, we can use the Lloyd-Max quantizer, which finds the optimal biased quantization for a given distribution. In particular, we get that the optimal (non-discrete) *vNMSE* is at least $0.35, 0.11, 0.031, 0.0082$ for $b = 1, 2, 3, 4$, accordingly, Compared to unbiased QUIC-FL's *vNMSE* of $1.52, 0.223, 0.044, 0.0098$. Note that as $b$ grows, QUIC-FL's *vNMSE* quickly approaches the Lloyd-Max lower bound for biased quantization.

## 4 EVALUATION

In this section, we evaluate the fully-fledged version of QUIC-FL that leverages RHT and client-specific shared randomness, as given in Appendix F and Algorithm 3.

**Parameter selection.** We experiment with how the different parameters (number of quantiles $m$, the fraction of coordinates sent exactly $p$, the number of shared random bits $\ell$, etc.) affect the performance of our algorithm. As shown in Figure 2, introducing shared randomness significantly decreases the *NMSE* compared with Algorithm 1 (i.e., $\ell = 0$). We note that these results are essentially independent of the input data (because of the RHT). Additionally, the benefit from adding each additional shared random bit diminishes, and the gain beyond $\ell = 4$ is negligible, especially for large $b$. Accordingly, we hereafter use $\ell = 6$ for $b = 1$, $\ell = 5$ for $b = 2$, and $\ell = 4$ for $b \in \{3, 4\}$. With respect to $p$, we determined $1/512$ as a good balance between the *NMSE* and bandwidth overhead for accurately sent values and their indices.

**Comparison to state-of-the-art DME techniques.** Next, we compare the performance of QUIC-FL to the baseline algorithms in terms of *NMSE*, encoding speed, and decoding speed, using an NVIDIA 3080 RTX GPU machine with 32GB RAM and i7-10700K CPU @ 3.80GHz. Specifically, we

compare with inputs where each coordinate is independently $LogNormal(0, 1)$ Chmiel et al. (2020). Hadamard Suresh et al. (2017), Kashin's representation Caldas et al. (2018); Safaryan et al. (2020), QSGD Alistarh et al. (2017), and EDEN Vargaftik et al. (2022). We evaluate two variants of Kashin's representation: (1) The TensorFlow (TF) implementation Google that, by default, limits the decomposition to three iterations, and (2) the theoretical algorithm that requires $O(\log(n \cdot d))$ iterations. For this experiment, the coordinates are As shown in Figure 3, QUIC-FL has significantly faster decoding than EDEN (as previously conveyed in Figure 1), the only alternative with competitive $NMSE$.

QUIC-FL is also significantly more accurate than all other approaches. We observe that the default TF configuration of Kashin's representation suffers from a bias, and therefore its $NMSE$ is not $O(1/n)$. In contrast, the theoretical algorithm is unbiased but has an asymptotically slower encoding time. We observed similar trends for different $n, b$, and $d$ values. We consider the algorithms' bandwidth over all coordinates (i.e., with $b + \frac{64}{512}$ bits for QUIC-FL, namely a float and a 32-bit index for each accurately sent entry). We evaluate the algorithms on additional input distributions and report similar results in Appendix H. Overall, the empirical measurements fall in line with the bounds in Table 1.

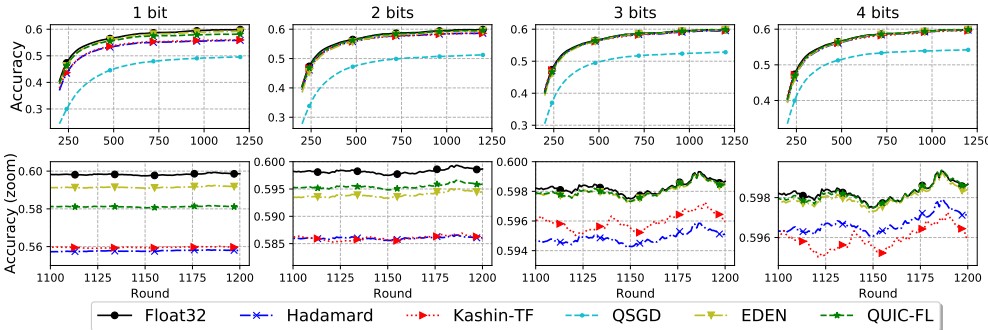

Figure 4: *FedAvg* over the Shakespeare next-word prediction task at various bit budgets (rows). We report training accuracy per round with a rolling mean of 200 rounds.

**Federated Learning Experiments.** We evaluate QUIC-FL over the Shakespeare next-word prediction task Shakespeare; McMahan et al. (2017) using an LSTM recurrent model. It was first suggested in McMahan et al. (2017) to naturally simulate a realistic heterogeneous federated learning setting. We run *FedAvg* McMahan et al. (2017) with the Adam server optimizer Kingma & Ba (2015) and sample $n = 10$ clients per round. We use the setup from the federated learning benchmark of Reddi et al. (2021), restated for convenience in Appendix I. Figure 4 shows how QUIC-FL is competitive with the asymptotically slower EDEN and markedly more accurate than other alternatives.

Due to space limits, experiments for image classification (Appendix J.1), a framework that uses DME as a building block (Appendix J.2), and power iteration (Appendix J.3), appear in the appendix.

## 5    RELATED WORKS

In Section 1, we gave an extensive overview of most related works, namely, other DME methods. In Appendix B, we give a broader overview of other compression and acceleration techniques, including frameworks that use DME as a building block; bounded support quantization alternatives; distribution-aware quantization; Entropy encoding techniques; methods that use client-side memory; error feedback solutions; opportunities in aggregating things other than gradients (such as gradient differences); in-network aggregation; sparsification approaches; shared randomness applications; non-uniform quantization; improvements by leveraging gradient correlations; and privacy concerns.

## 6    LIMITATIONS

We view the main limitation of QUIC-FL as its inability to leverage structure in the gradient (e.g., correlations across coordinates). While some structure (e.g., sparsity) is extractable (e.g., by encoding just the non-zero coordinates and separately encoding the coordinate positions that are zero), other types of structure may be ruined by applying RHT. For example, if all the coordinates are $\pm 1$, it is possible to send the gradient exactly using one bit per coordinate, while QUIC-FL would have a small error.

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

## A  ON EDEN AND DRIVE WITH RHT

EDEN Vargaftik et al. (2022) and DRIVE Vargaftik et al. (2021) are only proven to be unbiased when using uniform random rotation (which takes $\Theta(d^3)$ time). When using RHT, its quantization is biased and (if gradients are similar to each other) can have an *NMSE* that does not decay as a function of $n$. For example, consider DRIVE (or EDEN with $b = 1$), i.e., using the centroids $\pm 1/\sqrt{2}$ and the input $(1, 0.99, 0, 0, \ldots, 0)$. Both algorithms with RHT will estimate the vector as $(1, 0, 0, \ldots, 0)$ since $\texttt{sign}(HDx)$ is only determined by $D[0]$ and transformed coordinates are quantized to $1/\sqrt{2}$ if their sign is positive or $-1/\sqrt{2}$ otherwise. This means that the quantization is biased and that if all clients hold the above vector the *NMSE* would be $O(1)$ and not $O(1/n)$.

## B  EXTENDED RELATED WORK

This paper focused on the Distributed Mean Estimation (DME) problem where clients send lossily compressed vectors to a centralized server for averaging. While this problem is worthy of study on its own merits, we are particularly interested in applications to federated learning, where there are many variations and practical considerations, which have led to many alternative compression methods to be considered.

We note that in essence, QUIC-FL is a compression scheme. However, unlike previous DME approaches such as Suresh et al. (2017); Vargaftik et al. (2021; 2022), it brings benefits *only* in a distributed setting with multiple clients, distinguishing it from standard vector quantization methods.

**Frameworks that use DME as a building block.**  In addition to EF21 Richtárik et al. (2021) and MARINA Gorbunov et al. (2021); Szlendak et al. (2022) which are discussed in detail below, there are additional frameworks that leverage DME as a building block. For example, EF-BV Condat et al. (2022b), Qsparse-local-SGD Basu et al. (2019), 3PC Richtárik et al. (2022), CompressedScaffnew Condat et al. (2022a), MURANA Condat & Richtárik (2022), and DIANA Horváth et al. (2023) accelerate the convergence of non-convex learning tasks via variance reduction, control variates, and compression. These approaches are orthogonal and can benefit from better DME techniques such as QUIC-FL.

**Bounded support quantization.** Previous works on compression in federated learning observed considered bounding the range of the updates. They suggest ad-hoc mitigations, such as clipping Zhang et al. (2020); Wen et al. (2017); Zhang et al. (2022); Charles et al. (2021), preconditioning Suresh et al. (2017); Caldas et al. (2018), and bucketing Alistarh et al. (2017). On the other hand, methods such as Top-$k$ Stich et al. (2018a); Sinha et al. (2020) demonstrate that considering the largest coordinates is advantageous. Horváth & Richtarik (2021) provides convergence guarantees from combining biased and unbiased compressed estimators. BSQ similarly tries to benefit by sending the largest transformed coordinates exactly while sending the rest via unbiased compression.

We note that BSQ is also related to the threshold-$v$ algorithm Dutta et al. (2020) (for some $v > 0$) that sends accurately all the coordinates instead of $[-v, v]$. Namely, if we pick $v = t_p$ such that no more than $p$-fraction of the coordinates can fall outside $[-t_p, t_p]$, the algorithms coincide. There are some notable differences: first, we analyze the theoretical vNMSE of BSQ and show that it asymptotically improves the worst-case compared to without BSQ. Second, we use it in conjunction with RHT to obtain a bounded support distribution that we can optimize the quantization for using our solver.

**Distribution-aware quantization.** Quantization over a distribution, and over a Gaussian source in particular, has been studied for almost a century (for a comprehensive overview, we refer to Gray & Neuhoff (1998)). Nevertheless, to our knowledge, such research has not focused on the unbiasedness constraint. The only comparable methods that we are aware of are based on stochastic quantization and introduce an error that increases with the vector's dimension. There are additional unbiased methods that use shared randomness (e.g., Roberts (1962a); Ben Basat et al. (2021)), but again, we are unaware of any work that directly optimizes quantization for a distribution with an unbiasedness constraint. As previously mentioned, perhaps the closest to our approach is the Lloyd-Max Scalar Quantizer Lloyd (1982); Max (1960), which optimizes the mean squared error without unbiasedness constraints. Interestingly, there are many generalizations to Lloyd-Max, such as vector quantization Linde et al.

(1980) methods and lattice quantization Gersho (1979). In future work, we plan to investigate these approaches and extend our distribution-aware unbiasedness quantization framework accordingly.

**Entropy encoding.** When the encoding and decoding time is less important, some previous approaches have suggested using an entropy encoding such as Huffman or arithmetic encoding to improve the accuracy (e.g., Alistarh et al. (2017); Suresh et al. (2017); Vargaftik et al. (2022); Dorfman et al. (2023)). Intuitively, such encodings allow us to losslessly compress the lossily compressed vector to reduce its representation size, thereby allowing less aggressive quantization. However, we are unaware of available GPU-friendly entropy encoding implementation and thus such methods incur a significant time overhead.

**Client-side memory.** Critically, for the basic DME problem, the assumption is that this is a one-shot process where the goal is to optimize the accuracy without relying on client-side memory. This model naturally fits cross-device federated learning, where different clients are sampled in each round. We focused on unbiased compression, which is standard in prior works Suresh et al. (2017); Konečnỳ & Richtárik (2018); Vargaftik et al. (2021); Davies et al. (2021); Mitchell et al. (2022). However, if the compression error is low enough, and under some assumptions, SGD can be proven to converge even with biased compression Beznosikov et al. (2020).

**Error feedback.** In other settings, such as distributed learning or cross-silo federated learning, we may assume that clients are persistent and have a memory that keeps state between rounds. A prominent option to leverage such a state is to use Error Feedback (EF). In EF, clients can track the compression error and add it to the vector computed in the consecutive round. This scheme is often shown to recover the model's convergence rate and resulting accuracy Seide et al. (2014); Alistarh et al. (2018); Richtárik et al. (2021); Karimireddy et al. (2019) and enables biased compressors such as Top-$k$ Stich et al. (2018a) and SignSGD Bernstein et al. (2018). We compare with the state of the art technique, EF21 Richtárik et al. (2021), in addition to showing how it can be used in conjunction with QUIC-FL to facilitate further improvement in Appendix J.

**Gradient differences.** An orthogonal proposal that works with persistent clients, which is also applicable with QUIC-FL, is to encode the difference between the current vector and the previous one instead of directly compressing the vector Mishchenko et al. (2019); Gorbunov et al. (2021). Broadly speaking, this allows a compression error proportional to the L2 norm of the difference and not the vector and can decrease the error if consecutive vectors are similar to each other.

**In-network aggregation.** When running distributed learning in cluster settings, recent works show how in-network aggregation can accelerate the learning process Sapio et al. (2021); Lao et al. (2021); Segal et al. (2021); Li et al. (2023). IntSGD Mishchenko et al. (2022) is another compression scheme that allows one to aggregate the compressed integer vectors in the network. However, their solution may require sending $14$ bits per coordinate while we consider $1-5$ bits per coordinate in QUIC-FL. Intuitively, switches are designed to move data at high speeds, and recent advances in switch programmability enable them to easily perform simple aggregation operations like summation while processing the data. Extending QUIC-FL to allow efficient in-network aggregation is left as future work.

**Sparsification.** Another line of work focuses on sparsifying the vectors before compressing them Konečný et al. (2017); Aji & Heafield (2017); Konečnỳ & Richtárik (2018); Wangni et al. (2018); Stich et al. (2018b); Fei et al. (2021); Vargaftik et al. (2022). Intuitively, in some learning settings, many of the coordinates are small, and we can improve the accuracy to bandwidth tradeoff by removing all small coordinates prior to compression. Another form of sparsification is random sampling, which allows us to avoid sending the coordinate indices Konečný et al. (2017); Vargaftik et al. (2022). We note that combining such approaches with QUIC-FL is straightforward, as we can use QUIC-FL to compress just the non-zero entries of the sparsified vectors.

**Deep gradient compression.** By combining techniques like warm-up training, vector clipping, momentum factor masking, momentum correction, and deep vector compression, Lin et al. (2018) reports savings of two orders of magnitude in the bandwidth required for distributed learning.

**Shared randomness.** As shown in Ben Basat et al. (2021), shared randomness can reduce the worst-case error of quantizing a single $[0, 1]$ value both in biased and unbiased settings. However, applying this approach directly to the vector's entries results in $O(d/n)$ *NMSE* for any $b = O(1)$. Another promising orthogonal approach is to leverage shared randomness to push the clients' compression to yield errors in opposite directions, thus making them cancel out and lowering the overall NMSE Suresh et al. (2022); Szlendak et al. (2022).

**Non-uniform quantization.** The QUIC-FL algorithm, based on the output of the solver (see §3), uses *non-uniform* quantization, i.e., has quantization levels that are not uniformly spaced. Indeed, recent works observed that non-uniform quantization improves the estimation accuracy and accelerates the learning convergence Ramezani-Kebrya et al. (2019); Faghri et al. (2020).

Our algorithm significantly improves the worst-case error bound obtained by NUQSGD Ramezani-Kebrya et al. (2019), ALQ Faghri et al. (2020), and AMQ Faghri et al. (2020). Namely (see (Faghri et al., 2020, Section 1) and (Ramezani-Kebrya et al., 2019, Theorem 4)), for the parameter range $b = O(1)$ that we consider in this paper, the vNMSE of NUQSGD, ALQ, and AMQ is $O(\sqrt{d})$ while QUIC-FL's is $O(1)$. Indeed, these works showed the benefit of choosing non-uniform quantization levels and improved the $O(d)$ vNMSE of QSGD. Further, their vNMSEs match the $\Omega(\sqrt{d})$ lower bound for non-uniform stochastic quantization that applies to algorithms that select the quantization levels directly for the input vector. However, this lower bound does not apply when using preprocessing (e.g., RHT), bounding the support (e.g., BSQ), or utilizing shared randomness, which are the techniques that allowed us to drive the vNMSE to a small constant that is independent of $d$.

**Correlations.** Some techniques further reduce the error by leveraging potential correlations between coordinates Mitchell et al. (2022) or client vectors Davies et al. (2021); it is unclear how to combine these with QUIC-FL and we leave this for future work.

**Privacy concerns** Several works optimize the communication-accuracy tradeoff while also considering the privacy of clients' data. For example the authors of Chen et al. (2020) optimize the triple communication-accuracy-privacy tradeoff, while Gandikota et al. (2021) addresses the harder problem of compressing the gradients while maintaining differential privacy. Their results can be split into two groups: (1) algorithms that require $O(\log d)$ bits per coordinate to reach the $O(1/n)$ NMSE, and (2) an algorithm that needs $O_\epsilon(1)$ bits per coordinate (which hides functions of $\epsilon$) to reach an NMSE of $\frac{1}{n \cdot (1-\epsilon)}$. In particular, the vNMSE of this approach is always larger than that of QUIC-FL, even for $b = 1$.

**Spherical compression.** Spherical compression (SC) Albasyoni et al. (2020) is a highly accurate biased quantization method that draws random points on a unit sphere until one is $\epsilon$-close to the vector's direction; it then sends just the number of points needed and the server uses the same pseudo-random number generator seed to compute the estimate. The algorithm runs in time $O(d/\mathfrak{p})$, where $\mathfrak{p}$ is the probability that a sampled point is $\epsilon$-close to the input and satisfies $\mathfrak{p} = \frac{1}{2} F_{(d-1)/2, 1/2}(\alpha)$, where $F$ is the CDF of the Beta distribution and $\alpha$ is desired the *vNMSE* bound. Evaluating this expression shows that is excessively large when $d$ is not very small. For example, for $d = 100$, they would require over $10^{33}$ samples on average (while we consider $d$ in the millions). More generally, $1/\mathfrak{p} \geq (1/\alpha)^{d/2}$, thus the encoding and decoding complexities are exponential. This is implied by the lower bound of Safaryan et al. (2020). Finally, we note that QUIC-FL is unbiased while the SC algorithm is biased (and thus, its *NMSE* does not decrease linearly in $n$).

**Sparse dithering.** Sparse dithering is a compression method that is shown to be near-optimal in the sense that it requires at most constant factor more bandwidth than the lower bound for the same error rate. We compare with it in Appendix J.4.

**Natural compression** Natural Compression and Dithering Horvóth et al. (2022) are schemes optimized for processing speed by taking into consideration the representation of floating point values when designing the compression. However, In order to get constant *vNMSE*, they seem to require $O(d \log d)$ bits compared with $O(d)$ bits in QUIC-FL and their *vNMSE* is lower bounded by $1/8$, while QUIC-FL achieves a *vNMSE* of $\approx 0.0444, \approx 0.00982$ with 3 and 4 bits per coordinate.

We refer the reader to Konečný et al. (2017); Kairouz et al. (2019); Xu et al. (2020); Wang et al. (2021) for an extensive review of the current state of the art and challenges.

## C    ANALYSIS OF THE BOUNDED SUPPORT QUANTIZATION TECHNIQUE

In this appendix, we analyze the Bounded Support Quantization (BSQ) approach that sends all coordinates outside a range $[-t_p, t_p]$ exactly and performs a standard (i.e., uniform) stochastic quantization for the rest.

Let $p \in (0,1)$ and denote $t_p = \frac{\|\overline{x}\|_2}{\sqrt{d \cdot p}}$; notice that there can be at most $d \cdot p$ coordinates outside $[-t_p, t_p]$. Using $b$ bits, we split this range into $2^b - 1$ intervals of size $\frac{2t_p}{2^b - 1}$, meaning that each coordinate's expected squared error is at most $\left( \frac{2t_p}{2^b - 1} \right)^2 / 4$. The MSE of the algorithm is therefore bounded by

$$\mathbb{E}\left[ \left\| \overline{x} - \widehat{\overline{x}} \right\|_2^2 \right] = d \cdot \left( \frac{2t_p}{2^b - 1} \right)^2 / 4 = \frac{\|\overline{x}\|_2^2}{p \cdot \left( 2^b - 1 \right)^2}.$$

This gives the result

$$vNMSE \leq \frac{1}{p \cdot \left( 2^b - 1 \right)^2}.$$

Thus, as clients use independent randomness for the quantization, we have that

$$NMSE \leq \frac{1}{n \cdot p \cdot \left( 2^b - 1 \right)^2}.$$

Let $r$ be the representation length of each coordinate in the input vector (e.g., $r = 32$ for single-precision floats) and $i$ be the number of bits that represent a coordinate's index (e.g., $i = 32$, assuming $\log d \leq 32$). Then, we get that BSQ sends a message with less than $p \cdot (r + i) + b$ bits per coordinate. Further, this method has $O(d)$ time for encoding and decoding and is GPU-friendly.

As mentioned in Section 3.2, it is possible to encode the indices of the exactly sent coordinates using only $\log \binom{d}{d \cdot p}$ bits at the cost of additional complexity. Also, it is possible to send a bit vector to indicate whether each coordinate is exactly sent or quantized and obtain a message with fewer than $p \cdot r + b + 1$ bits.

However, empirically we find the method of transmitting the indices without encoding most useful as $p \cdot \log d \ll 1$ in our settings, resulting in fast processing time and small bandwidth overhead.

## D    ON THE DISTRIBUTION OF ROTATED VECTORS

While our framework does not depend on any particular distribution of the input vectors, as we have noted we can apply pre-processing by applying a random rotation so that each coordinate is provably approximately normally distributed, and design once a near-optimal table for that case.

We further emphasize that the results obtained in this paper (Theorem G.3 and Theorem G.2) hold *when using RHT and for **any** input vector as well when using the same near-optimal tables designed for the uniform rotation* (as they consider the actual resulting distribution).

We discuss here the relevant theory of random rotations and RHT that form the basis for these results.

As analyzed by (Vargaftik et al., 2021, Appendix A.4), after a uniform random rotation, the coordinates follow a "shifted Beta" distribution, namely, if $Y \sim Beta(\frac{1}{2}, \frac{d-1}{2})$, then the distribution of each coordinate is identical to that of: $\frac{Y+1}{2}$. Next, it is known that this distribution quickly approaches that of a normal distribution when $d$ grows. Namely, if $X_n \sim Beta(\alpha n, \beta n)$ then $\sqrt{n} \left( X_n - \frac{\alpha}{\alpha + \beta} \right)$ converges to a normal random variable with mean 0 and variance $\frac{\alpha \beta}{(\alpha + \beta)^3}$ as $n$ increases.

With RHT, the resulting distribution slightly differs from the above. However, as proved in (Vargaftik et al., 2021, Section 6.2), it remains very similar under reasonable assumptions about the distribution

of the input vector, with the first five moments (and all odd ones) matching that of a normal distribution.

Again, the RHT-related results of Theorem G.3 and Theorem G.2 do not rely on this analysis of the transformed coordinate distribution.

# E    QUIC-FL'S *NMSE* PROOF

In this appendix, we analyze the *vNMSE* and then the *NMSE* of our algorithm.

Let $\chi = \mathbb{E}[(Z - \widehat{Z})^2]$ denote the error of the quantization of a normal random variable $Z \sim \mathcal{N}(0, 1)$. Our analysis is general and covers QUIC-FL, but is also applicable to any unbiased quantization method that is used following a uniform random rotation preprocessing.

Essentially, we show that QUIC-FL's *vNMSE* is $\chi$ plus a small additional additive error term (arising because the rotation does not yield exactly normally distributed and independent coordinates) that quickly tends to $0$ as the dimension increases.

**Lemma E.1.** *For QUIC-FL, it holds that:*

$$vNMSE \leq \chi + O\left(\sqrt{\frac{\log d}{d}}\right) .$$

*Proof.* The proof follows similar lines to that of Vargaftik et al. (2021; 2022). However, here the *vNMSE* expression is different and is somewhat simpler as it takes advantage of our *unbiased* quantization technique.

A rotation preserves a vector's euclidean norm. Thus, according to Algorithms 1 and 3 it holds that

$$\left\|\overline{x} - \widehat{\overline{x}}\right\|_2^2 = \left\|T\left(\overline{x} - \widehat{\overline{x}}\right)\right\|_2^2 = \left\|T\left(\overline{x}\right) - T\left(\widehat{\overline{x}}\right)\right\|_2^2 =$$
$$\left\|\frac{\|\overline{x}\|_2}{\sqrt{d}} \cdot \overline{Z} - \frac{\|\overline{x}\|_2}{\sqrt{d}} \cdot \widehat{\overline{Z}}\right\|_2^2 = \frac{\|\overline{x}\|_2^2}{d} \cdot \left\|\overline{Z} - \widehat{\overline{Z}}\right\|_2^2 . \tag{1}$$

Taking expectation and dividing by $\|\overline{x}\|_2^2$ yields

$$vNMSE \triangleq \mathbb{E}\left[\frac{\left\|\overline{x} - \widehat{\overline{x}}\right\|_2^2}{\|\overline{x}\|_2^2}\right] = \frac{1}{d} \cdot \mathbb{E}\left[\left\|\overline{Z} - \widehat{\overline{Z}}\right\|_2^2\right]$$

$$= \frac{1}{d} \cdot \mathbb{E}\left[\sum_{i=0}^{d-1}\left(\overline{Z}[i] - \widehat{\overline{Z}}[i]\right)^2\right] = \frac{1}{d} \cdot \sum_{i=0}^{d-1} \mathbb{E}\left[\left(\overline{Z}[i] - \widehat{\overline{Z}}[i]\right)^2\right] . \tag{2}$$

Let $\widetilde{\overline{Z}}$ be a vector of $d$ independent $\mathcal{N}(0, 1)$ random variables. Then the distribution of each transformed and scaled coordinate $\overline{Z}[i]$ is given by $\overline{Z}[i] \sim \sqrt{d} \cdot \frac{\widetilde{\overline{Z}}[i]}{\left\|\widetilde{\overline{Z}}\right\|_2}$ (e.g., see Vargaftik et al. (2021); Muller (1959)).

This means that all coordinates of $\overline{Z}$ follow the same distribution, and thus all coordinates of $\widehat{\overline{Z}}$ follow the same (different) distribution. Thus, without loss of generality, we obtain

$$vNMSE \triangleq \mathbb{E}\left[\frac{\left\|\overline{x} - \widehat{\overline{x}}\right\|_2^2}{\|\overline{x}\|_2^2}\right] = \mathbb{E}\left[\left(\overline{Z}[0] - \widehat{\overline{Z}}[0]\right)^2\right] = \mathbb{E}\left[\left(\frac{\sqrt{d}}{\left\|\widetilde{\overline{Z}}\right\|_2} \cdot \widetilde{\overline{Z}}[0] - \widehat{\overline{Z}}[0]\right)^2\right] . \tag{3}$$

For some $0 < \alpha < \frac{1}{2}$, denote the event

$$\mathscr{E} = \left\{d \cdot (1 - \alpha) \leq \left\|\widetilde{\overline{Z}}\right\|_2^2 \leq d \cdot (1 + \alpha)\right\} .$$

Let $\mathscr{E}^c$ be the complementary event of $\mathscr{E}$. By Lemma D.2 in Vargaftik et al. (2022) it holds that $\Pr[\mathscr{E}^c] \le 2 \cdot e^{-\frac{\alpha^2}{8} \cdot d}$. Also, by the law of total expectation

$$
\mathbb{E}\left[ \left( \frac{\sqrt{d}}{\left\| \overline{\widetilde{Z}} \right\|_2} \cdot \overline{\widetilde{Z}}[0] - \widehat{\overline{\widetilde{Z}}}[0] \right)^2 \right] \le
$$

$$
\mathbb{E}\left[ \left( \frac{\sqrt{d}}{\left\| \overline{\widetilde{Z}} \right\|_2} \cdot \overline{\widetilde{Z}}[0] - \widehat{\overline{\widetilde{Z}}}[0] \right)^2 \middle| \mathscr{E} \right] \cdot \Pr[\mathscr{E}] + \mathbb{E}\left[ \left( \frac{\sqrt{d}}{\left\| \overline{\widetilde{Z}} \right\|_2} \cdot \overline{\widetilde{Z}}[0] - \widehat{\overline{\widetilde{Z}}}[0] \right)^2 \middle| \mathscr{E}^c \right] \cdot \Pr[\mathscr{E}^c] \le \quad (4)
$$

$$
\mathbb{E}\left[ \left( \frac{\sqrt{d}}{\left\| \overline{\widetilde{Z}} \right\|_2} \cdot \overline{\widetilde{Z}}[0] - \widehat{\overline{\widetilde{Z}}}[0] \right)^2 \middle| \mathscr{E} \right] \cdot \Pr[\mathscr{E}] + M \cdot \Pr[\mathscr{E}^c] ,
$$

where $M = (vNMSE_{\max})^2$ and $vNMSE_{\max}$ is the maximal value that the server can reconstruct (i.e., $\max(Q_{b,p})$ in Algorithm 1 or $\max(R)$ in Algorithm 3) which is a constant that is *independent* of the vector's dimension. Next,

$$
\mathbb{E}\left[ \left( \frac{\sqrt{d}}{\left\| \overline{\widetilde{Z}} \right\|_2} \cdot \overline{\widetilde{Z}}[0] - \widehat{\overline{\widetilde{Z}}}[0] \right)^2 \middle| \mathscr{E} \right] = \mathbb{E}\left[ \left( \left( \overline{\widetilde{Z}}[0] - \widehat{\overline{\widetilde{Z}}}[0] \right) + \left( \frac{\sqrt{d}}{\left\| \overline{\widetilde{Z}} \right\|_2} - 1 \right) \cdot \overline{\widetilde{Z}}[0] \right)^2 \middle| \mathscr{E} \right] =
$$

$$
\mathbb{E}\left[ \left( \overline{\widetilde{Z}}[0] - \widehat{\overline{\widetilde{Z}}}[0] \right)^2 \middle| \mathscr{E} \right] + 2 \cdot \mathbb{E}\left[ \left( \overline{\widetilde{Z}}[0] - \widehat{\overline{\widetilde{Z}}}[0] \right) \cdot \left( \frac{\sqrt{d}}{\left\| \overline{\widetilde{Z}} \right\|_2} - 1 \right) \cdot \overline{\widetilde{Z}}[0] \middle| \mathscr{E} \right] + \quad (5)
$$

$$
\mathbb{E}\left[ \left( \left( \frac{\sqrt{d}}{\left\| \overline{\widetilde{Z}} \right\|_2} - 1 \right) \cdot \overline{\widetilde{Z}}[0] \right)^2 \middle| \mathscr{E} \right]
$$

Also,

$$
\mathbb{E}\left[ \left( \overline{\widetilde{Z}}[0] - \widehat{\overline{\widetilde{Z}}}[0] \right) \cdot \left( \frac{\sqrt{d}}{\left\| \overline{\widetilde{Z}} \right\|_2} - 1 \right) \cdot \overline{\widetilde{Z}}[0] \middle| \mathscr{E} \right] \cdot \Pr[\mathscr{E}] \le
$$

$$
\left( \frac{1}{\sqrt{1-\alpha}} - 1 \right) \cdot \left| \mathbb{E}\left[ \left( \overline{\widetilde{Z}}[0] - \widehat{\overline{\widetilde{Z}}}[0] \right) \cdot \overline{\widetilde{Z}}[0] \middle| \mathscr{E} \right] \cdot \Pr[\mathscr{E}] \right| \le \quad (6)
$$

$$
\left( \frac{1}{\sqrt{1-\alpha}} - 1 \right) \cdot \left| \mathbb{E}\left[ \left( \overline{\widetilde{Z}}[0] \right)^2 - \widehat{\overline{\widetilde{Z}}}[0] \cdot \overline{\widetilde{Z}}[0] \middle| \mathscr{E} \right] \cdot \Pr[\mathscr{E}] \right| \le
$$

$$
\left( \frac{1}{\sqrt{1-\alpha}} - 1 \right) \cdot 1 + \left( \frac{1}{\sqrt{1-\alpha}} - 1 \right) \cdot \frac{1}{\sqrt{1-\alpha}} = \frac{\alpha}{1-\alpha} \le 2\alpha .
$$

Here, we used that

$$
\mathbb{E}\left[ \left( \overline{\widetilde{Z}}[0] \right)^2 \middle| \mathscr{E} \right] \cdot \Pr[\mathscr{E}] \le \mathbb{E}\left[ \left( \overline{\widetilde{Z}}[0] \right)^2 \right] = 1 ,
$$

and that

$$
\mathbb{E}\left[ \widehat{\overline{\widetilde{Z}}}[0] \cdot \overline{\widetilde{Z}}[0] \middle| \mathscr{E} \right] \cdot \Pr[\mathscr{E}] = \mathbb{E}\left[ \mathbb{E}\left[ \widehat{\overline{\widetilde{Z}}}[0] \cdot \overline{\widetilde{Z}}[0] \middle| \mathscr{E}, \overline{\widetilde{Z}} \right] \right] \cdot \Pr[\mathscr{E}]
$$

$$
= \mathbb{E}\left[ \frac{\sqrt{d}}{\left\| \overline{\widetilde{Z}} \right\|_2} \cdot \left( \overline{\widetilde{Z}}[0] \right)^2 \middle| \mathscr{E} \right] \cdot \Pr[\mathscr{E}] \le \frac{1}{\sqrt{1-\alpha}} \cdot \mathbb{E}\left[ \left( \overline{\widetilde{Z}}[0] \right)^2 \right] = \frac{1}{\sqrt{1-\alpha}}. \quad (7)
$$

Next, we similarly obtain

$$\mathbb{E}\left[\left(\left(\frac{\sqrt{d}}{\left\|\overline{\widetilde{Z}}\right\|_2}-1\right)\cdot\overline{\widetilde{Z}}[0]\right)^2\middle|\mathscr{E}\right]\cdot\Pr[\mathscr{E}]\le\quad\left(\frac{1}{\sqrt{1-\alpha}}-1\right)+\left(1-\frac{1}{\sqrt{1+\alpha}}\right)\le 2\alpha.\quad(8)$$

Thus,

$$vNMSE\le\mathbb{E}\left[\left(\overline{\widetilde{Z}}[0]-\widehat{\overline{\widetilde{Z}}}[0]\right)^2\right]+4\alpha+2\cdot e^{-\frac{\alpha^2}{8}\cdot d}\cdot M\,.\quad(9)$$

Setting $\alpha=\sqrt{\frac{8\log d}{d}}$ yields $vNMSE\le\mathbb{E}\left[\left(\overline{\widetilde{Z}}[0]-\widehat{\overline{\widetilde{Z}}}[0]\right)^2\right]+O\left(\sqrt{\frac{\log d}{d}}\right)$.

Since $\overline{\widetilde{Z}}[0]\sim\mathcal{N}(0,1)$, we can write

$$vNMSE\le\mathbb{E}\left[\left(Z-\widehat{Z}\right)^2\right]+O\left(\sqrt{\frac{\log d}{d}}\right)\,.$$

This concludes the proof of the Lemma. $\qquad\square$

We are now ready to prove the theorem.

**Theorem 3.1.** *Let $Z\sim\mathcal{N}(0,1)$ and let $\widehat{Z}$ be its estimation by our distribution-aware unbiased quantization scheme. Then, for any number of clients $n$ and any set of $d$-dimensional input vectors $\left\{\overline{x}_c\in\mathbb{R}^d\mid c\in\{0,\dots,n-1\}\right\}$, we have that QUIC-FL's NMSE respects*

$$NMSE=\frac{1}{n}\cdot\mathbb{E}\left[\left(Z-\widehat{Z}\right)^2\right]+O\left(\frac{1}{n}\cdot\sqrt{\frac{\log d}{d}}\right).$$

*Proof.* We start by analyzing QUIC-FL's $\chi$. We can write:

$$\chi=\mathbb{E}\left[\left(Z-\widehat{Z}\right)^2\right]=\mathbb{E}\left[\left(Z-\widehat{Z}\right)^2\mid Z\in[-t_p,t_p]\right]\cdot\Pr[Z\in[-t_p,t_p]]\quad+$$
$$\mathbb{E}\left[\left(Z-\widehat{Z}\right)^2\mid Z\notin[-t_p,t_p]\right]\cdot\Pr[Z\notin[-t_p,t_p]],\quad(10)$$

where the first summand is exactly the quantization error of our distribution-aware unbiased BSQ, and the second summand is $0$ as such values are sent exactly.

This means that for any $b$ and $p$, we can exactly compute $\chi$ given the solver's output (i.e., the precomputed quantization-values or tables). For example, it is $\approx 8.58$ for $b=1,\ell=0$ and $p=2^{-9}$.

By Lemma E.1, we get that QUIC-FL's $vNMSE$ is $\chi+O\left(\sqrt{\frac{\log d}{d}}\right)=O(1)$.

Since the clients' quantization is independent, we immediately obtain the result as $NMSE=\frac{1}{n}\cdot vNMSE$. $\qquad\square$

## F  QUIC-FL WITH CLIENT-SPECIFIC SHARED RANDOMNESS

In the most general problem formulation, we assume that the sender and receiver have access to a shared $h\sim U[0,1]$ random variable. This corresponds to having infinite shared random bits. Using this shared randomness, for each message $x\in\mathcal{X}_b$, the sending client chooses the probability $S(h,z,x)$ to quantize its value $z\in[-t_p,t_p]$ to the associated value $R(h,x)$ reconstructed by the receiver. We emphasize that $h$ does not need to be transmitted. We further note that the unbiasedness constraint is now defined with respect to both the private randomness of the client (which is used to pick a message with respect to the distribution $S$) and the (client-specific) shared randomness $h$. This yields the following optimization problem:

$$\underset{S,R}{\text{minimize}} \qquad \int_0^1 \int_{-t_p}^{t_p} \sum_{x \in \mathcal{X}_b} S(h,z,x) \cdot (z - R(h,x))^2 \cdot e^{\frac{-z^2}{2}} \, dz \, dh$$

subject to

$$(\textit{Unbiasedness}) \quad \int_0^1 \sum_{x \in \mathcal{X}_b} S(h,z,x) \cdot R(h,x) \, dh = z, \qquad\qquad \forall z \in [-t_p, t_p]$$

$$(\textit{Probability}) \qquad \sum_{x \in \mathcal{X}_b} S(h,z,x) = 1, \qquad\qquad\qquad \forall h \in [0,1], \, z \in [-t_p, t_p]$$

$$S(h,z,x) \ge 0, \qquad\qquad\qquad \forall h \in [0,1], \, z \in [-t_p, t_p], \, x \in \mathcal{X}_b$$

As in the case without shared randomness, we are unaware of analytical methods for solving this continuous problem. Therefore, we discretize it to get a problem with finitely many variables. To that end, we further discretize the client-specific shared randomness, allowing $h \in \mathcal{H}_\ell = \{0, \ldots, 2^\ell - 1\}$ to have $\ell$ shared random bits. As with the number of quantiles $m$, the parameter $\ell$ gives a tradeoff between the complexity of the resulting (discretized) problem and the error of the quantization.

---

**Algorithm 2** QUIC-FL with client-specific shared randomness and stoch. quantizing to quantiles

---

**Input:** Bit budget $b$, shared random bits $\ell$, BSQ parameter $p$ and its threshold $t_p$ and precomputed quantiles $\mathcal{A}_{p,m}$, sender table $S$ and receiver table $R$.

---

**Client $c$:**

1. $\overline{Z}_c \leftarrow \frac{\sqrt{d}}{\|\overline{x}_c\|_2} \cdot T\left(\overline{x}_c\right)$

2. $\overline{U}_c, \overline{I}_c \leftarrow \left\{\overline{Z}_c[i] \,\middle|\, \left|\overline{Z}_c[i]\right| > t_p\right\}, \left\{i \,\middle|\, \left|\overline{Z}_c[i]\right| > t_p\right\}$

3. $\overline{V}_c \leftarrow \left\{z \in \overline{Z}_c \,\middle|\, |z| \le t_p\right\}$

4. $\widetilde{\overline{V}}_c \leftarrow$ Stochastically quantize $\overline{V}_c$ using $\mathcal{A}_{p,m}$

5. $\overline{H}_c \leftarrow \left\{\forall i : \text{Sample } \overline{H}_c[i] \sim \mathcal{U}[\mathcal{H}_\ell]\right\}$

6. $\overline{X}_c \leftarrow \left\{\forall i : \text{Sample } \overline{X}_c[i] \sim \left\{x \text{ with prob. } S(\overline{H}_c[i], \widetilde{\overline{V}}_c[i], x) \mid x \in \mathcal{X}_b\right\}\right\}$

7. Send $\left(\|\overline{x}_c\|_2, \overline{X}_c, \overline{U}_c, \overline{I}_c\right)$ to server

---

**Server:**

8.  For all $c$:

9.     $\overline{H}_c \leftarrow \left\{\forall i : \text{Sample } \overline{H}_c[i] \sim \mathcal{U}[\mathcal{H}_\ell]\right\}$

10.    $\widehat{\overline{V}}_c \leftarrow \left\{\forall i : R(\overline{H}_c[i], \overline{X}_c[i])\right\}$

11.    $\widehat{\overline{Z}}_c \leftarrow$ Merge $\widehat{\overline{V}}_c$ and $\left(\overline{U}_c, \overline{I}_c\right)$

12. $\widehat{\overline{Z}}_{avg} \leftarrow \frac{1}{n} \cdot \sum_{c=0}^{n-1} \frac{\|\overline{x}_c\|_2}{\sqrt{d}} \cdot \widehat{\overline{Z}}_c$

13. $\widehat{\overline{x}}_{avg} \leftarrow T^{-1}\left(\widehat{\overline{Z}}_{avg}\right)$

---

We give the formulation below (with the differences from the no-client-specific-shared-randomness version highlighted in red.)

$$\underset{S', R}{\text{minimize}} \qquad \sum_{\substack{h \in \mathcal{H}_\ell \\ i \in \mathcal{I}_m \\ x \in \mathcal{X}_b}} S'(h, i, x) \cdot (\mathcal{A}_{p,m}(i) - R(h, x))^2$$

subject to

$$(\textit{Unbiasedness}) \quad \frac{1}{2^\ell} \cdot \sum_{\substack{h \in \mathcal{H}_\ell \\ x \in \mathcal{X}_b}} S'(h, i, x) \cdot R(h, x) = \mathcal{A}_{p,m}(i), \qquad \forall i \in \mathcal{I}_m$$

$$(\textit{Probability}) \quad \sum_{x \in \mathcal{X}_b} S'(h, i, x) = 1, \qquad \forall h \in \mathcal{H}_\ell, i \in \mathcal{I}_m$$

$$S'(h, i, x) \ge 0, \qquad \forall h \in \mathcal{H}_\ell, i \in \mathcal{I}_m, x \in \mathcal{X}_b$$

Unlike without client-specific shared randomness, the solver's output does not directly yield an implementable algorithm, as it only associates probabilities to each $\langle h, i, x \rangle$ tuple. A natural option is to first stochastically quantize every rotated coordinate $Z \in [-t_p, t_p]$ to a one of the two closest quantiles before running the algorithm that is derived from solving the discrete optimization problem. The resulting pseudocode is shown in Algorithm 2.

The resulting algorithm is near-optimal in the sense that as the number of quantiles and shared random bits tend to infinity, we converge to an optimal algorithm. In practice, the solver is only able to produce an output for finite $m, \ell$ values; this means that the algorithm would be optimal if coordinates are uniformly distributed over $\mathcal{A}_{p,m}$.

In words Algorithm 2 starts similarly to Algorithm 1 by transforming and scaling the vector before splitting it to the large coordinates (that are sent accurately along with their indices) and the small coordinates (that are to be quantized). The difference is in the quantization process; Algorithm 2 first stochastically quantizes each small coordinate to a quantile in $\mathcal{A}_{p,m}$. Next, the client generates the (client-specific) shared randomness $\overline{H}_c$ and uses the pre-computed table $S$ to sample a message for each coordinate. That is, for each coordinate $i$, knowing the shared random value $\overline{H}_c[i]$ and the (rounded-to-quantile) transformed coordinate $\widetilde{\overline{V}}_c[i]$, for all $x \in \mathcal{X}_b$, $S(\overline{H}_c[i], \widetilde{\overline{V}}_c[i], x)$ is the probability that the client should send the message $x$. We note that the message for the $i$'th coordinate is sampled from $x$ w.p. $S(\overline{H}_c[i], \widetilde{\overline{V}}_c[i], x)$ using the client's private randomness. Finally, the client sends its vector's norm, the sampled messages, and the values and indices of the large transformed coordinates.

In turn, the server's algorithm is also similar to Algorithm 1, except for the estimation of the small transformed coordinates. In particular, for each client $c$, the server generates the client-specific shared randomness $\overline{H}_c$ and uses it to estimate each transformed coordinate $i$ using $R(\overline{H}_c[i], \overline{X}_c[i])$.

### F.1 Interpolating the Solver's Solution

A different approach, based on our examination of solver outputs, to yield an implementable algorithm from the optimal solution to the discrete problem is to calculate the message distribution directly from the rotated values without stochastically quantizing as we do in Algorithm 2. Indeed, we have found this approach somewhat faster and more accurate.

A crucial ingredient in getting a human-readable solution from the solver is that we, without loss of generality, force monotonicity in both $h$ and $x$, i.e., $(x \geq x') \wedge (h \geq h') \implies R(h, x) \geq R(h', x')$. We further found symmetry in the optimal sender and receiver tables for small values of $\ell$ and $m$. We then forced this symmetry to reduce the complexity of the solver's optimization problem size for larger $\ell$ and $m$ values. We use this symmetry in our interpolation.

**Examples, intuition and pseudocode.** We first explain the process by considering an example. We consider the setting of $p = \frac{1}{512}$ ($t_p \approx 3.097$), $m = 512$ quantiles, $b = 2$ bits per coordinate, and $\ell = 2$ bits of shared randomness. The solver's solution for the server's table $R$ is given below:

|  | $x = 0$ | $x = 1$ | $x = 2$ | $x = 3$ |
|---|---|---|---|---|
| $h = 0$ | -5.48 | -1.23 | **0.164** | 1.68 |
| $h = 1$ | -3.04 | -0.831 | **0.490** | 2.18 |
| $h = 2$ | -2.18 | **-0.490** | 0.831 | 3.04 |
| $h = 3$ | -1.68 | **-0.164** | 1.23 | 5.48 |

Table 2: Optimal server values ($R(h, x)$) for $x \in \mathcal{X}_2, h \in \mathcal{H}_2$ when $p = 1/512$ and $m = 512$, rounded to 3 significant digits.

The way to interpret the table is that if the server receives a message $x$ and the shared random value was $h$, it should estimate the (quantized) coordinate value as $R(h, x)$. For example, if $x = h = 2$, the estimated value would be $0.831$. We now explain what the table means for the sending client, starting with an example.

Consider $\overline{V}_c[i] = 0$. The question is: what message distribution should the sender use, given that $\overline{V}_c[i] \notin \mathcal{A}_{p,m}$ (and without quantizing the value to a quantile)? Based on the shared randomness value, we can use

$$\overline{X}_c[i] = \begin{cases} 1 & \text{If } \overline{H}_c[i] > 1 \\ 2 & \text{Otherwise} \end{cases}.$$

Indeed, we have that the estimate is unbiased as the receiver will estimate one of the bold entries in Table 2 with equal probabilities, i.e., $\mathbb{E}\left[\widehat{\overline{V}}_c[i]\right] = \frac{1}{4} \sum_{\overline{H}_c[i]} R(\overline{H}_c[i], \overline{X}_c[i]) = 0$.

Now, suppose that $\overline{V}_c[i] \in (0, t_p]$ (the case $\overline{V}_c[i] \in [-t_p, 0)$ is symmetric). The client can increase the server estimate's expected value (compared with the above choice of $\overline{X}_c[i]$'s distribution for

$\overline{V}_c[i] = 0$) by moving probability mass to larger $\overline{X}_c[i]$ values for some (or all) of the options for $\overline{X}_c[i]$.

For any $\overline{V}_c[i] \in (-t_p, t_p)$, there are infinitely many client alternatives that would yield an unbiased estimate. For example, if $\overline{V}_c[i] = 0.1$, below are two client options (rounded to three significant digit):

$$S_1(\overline{H}_c[i], \overline{V}_c[i], \overline{X}_c[i]) \approx \begin{cases} 1 & \text{If } (\overline{X}_c[i] = 1 \wedge \overline{H}_c[i] \leq 2) \\ 0.595 & \text{If } (\overline{X}_c[i] = 2 \wedge \overline{H}_c[i] = 3) \\ 0.405 & \text{If } (\overline{X}_c[i] = 3 \wedge \overline{H}_c[i] = 3) \\ 0 & \text{Otherwise} \end{cases}$$

$$S_2(\overline{H}_c[i], \overline{V}_c[i], \overline{X}_c[i]) \approx \begin{cases} 1 & \text{If } (\overline{X}_c[i] = 2 \wedge \overline{H}_c[i] \leq 1) \vee (\overline{X}_c[i] = 1 \wedge \overline{H}_c[i] = 3) \\ 0.697 & \text{If } (\overline{X}_c[i] = 1 \wedge \overline{H}_c[i] = 2) \\ 0.303 & \text{If } (\overline{X}_c[i] = 2 \wedge \overline{H}_c[i] = 2) \\ 0 & \text{Otherwise} \end{cases}$$

Note that while both $S_1$ and $S_2$ produce unbiased estimates, their expected squared errors differ. Further, since $0.1 \notin \mathcal{A}_{p,m}$, the solver's output does not directly indicate what is the optimal message distribution, even though the server table is known.

The approach we take corresponds to the following process. We move probability mass from the *leftmost, then uppermost* entry with non-zero mass to its right neighbor in the server table. So, for example, in Table 2, as $\overline{V}_c[i]$ increases from 0, we first move mass from the entry $\overline{H}_c[i] = 2, \overline{X}_c[i] = 1$ to the entry $\overline{H}_c[i] = 2, \overline{X}_c[i] = 2$. That is, the client, based on its private randomness, increases the probability of message $\overline{X}_c[i] = 2$ and decreases the probability of message $\overline{X}_c[i] = 1$ when $\overline{H}_c[i] = 2$. The amount of mass moved is always chosen to maintain unbiasedness. At some point, as $\overline{V}_c[i]$ increases, all of the probability mass will have moved, and then we start moving mass from $\overline{H}_c[i] = 3, \overline{X}_c[i] = 1$ similarly. (And subsequently, from $\overline{H}_c[i] = 0, \overline{X}_c[i] = 2$ and so on.)

This process is visualized in Figure 5. Note that $S(\overline{H}_c[i], \overline{V}_c[i], \overline{X}_c[i])$ values are piecewise linear as a function of $\overline{V}_c[i]$, and further, these values either go from 0 to 1, 1 to 0, or 0 to 1 and back again (all of which follow from our description). We can turn this description into formulae as explained below.

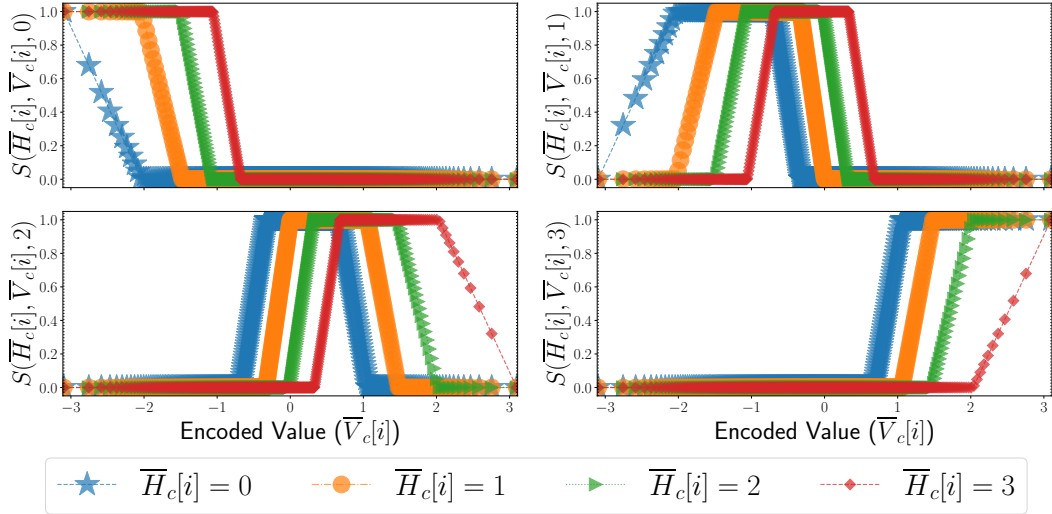

Figure 5: The interpolated solver's client algorithm for $b = \ell = 2, m = 512, p = \frac{1}{512}$. Markers correspond to quantiles in $\mathcal{A}_{p,m}$, and the lines illustrate our interpolation.

**Derivation of the interpolation equations.** We have found, by applying the mentioned monotonicity constraints (i.e., $(x \geq x') \wedge (h \geq h') \implies R(h, x) \geq R(h', x')$) and examining the solver's solutions for our parameter range, that the optimal approach for the client has a structure that we can generalize beyond specific examples. Namely, when the server table is monotone, the optimal solution *deterministically* quantizes the message to send in all but (at most) one shared randomness value. For instance, $S_2$ in the example above deterministically quantizes the message if $\overline{H}_c[i] \neq 2$ (sending $\overline{X}_c[i] = 1$ if $\overline{H}_c[i] = 3$ or $\overline{X}_c[i] = 2$ if $\overline{H}_c[i] \in \{0, 1\}$), or stochastically quantizes between $\overline{X}_c[i] = 1$ and $\overline{X}_c[i] = 2$ when $\overline{H}_c[i] = 2$. Furthermore, the shared randomness value in which we should stochastically quantize the message is easy to calculate.

To capture this behavior, we define the following quantities:

- The minimal message $\overline{X}_c[i]$ the client may send for $\overline{V}_c[i]$:

$$\underline{x}(\overline{V}_c[i]) = \max \left\{ x \in \mathcal{X}_b \;\middle|\; \left( \frac{1}{2^\ell} \cdot \sum_{\overline{H}_c[i] \in \mathcal{H}_\ell} R(\overline{H}_c[i], x) \right) \leq \overline{V}_c[i] \right\}.$$

That is, $\underline{x}(\overline{V}_c[i])$ is the maximal value such that sending $\underline{x}(\overline{V}_c[i])$ regardless of the shared randomness value would result in not overestimating $\overline{V}_c[i]$ in expectation. For example, as illustrated in Table 2 ($b = \ell = 2$), we have $\underline{x}(0) = 1$, as the client sends either 1 or 2 (highlighted in bold) depending on the shared randomness value.

- For convenience, we denote $R(h, 2^b) = \infty$ for all $h \in \mathcal{H}_\ell$. Then, the shared randomness value for which the sender stochastically quantizes is given by:

$$\underline{h}(\overline{V}_c[i]) = \max \left\{ h \in \mathcal{H}_\ell \;\middle|\; \frac{1}{2^\ell} \cdot \left( \sum_{h'=0}^{h-1} R(h', \underline{x}(\overline{V}_c[i]) + 1) + \sum_{h'=h}^{2^\ell - 1} R(h', \underline{x}(\overline{V}_c[i])) \right) \leq \overline{V}_c[i] \right\}.$$

That is, $\underline{h}(\overline{V}_c[i])$ denotes the maximal value for which sending $\left( \underline{x}(\overline{V}_c[i]) + 1 \right)$ if $\overline{H}_c[i] < \underline{h}(\overline{V}_c[i])$ or $\underline{x}(\overline{V}_c[i])$ if $\overline{H}_c[i] \geq \underline{h}(\overline{V}_c[i])$ would not overestimate $\overline{V}_c[i]$ in expectation. In the same example of Table 2 ($b = \ell = 2$), we have $\underline{h}(0) = 2$ since sending $\overline{X}_c[i] = 2$ for $h \leq 2$ would result in an overestimation.

**The sender-interpolated algorithm.** Let us denote by $\mu$ the expectation we require for $\overline{H}_c[i] = \underline{h}(\overline{V}_c[i])$ to ensure that our algorithm is unbiased:

$$\overline{\mu}_c[i] \triangleq \mathbb{E}\left[ \widehat{\overline{V}}_c[i] \;\middle|\; \overline{H}_c[i] = \underline{h}(\overline{V}_c[i]) \right] =$$

$$2^\ell \cdot \overline{V}_c[i] - \sum_{h=0}^{\underline{h}(\overline{V}_c[i]) - 1} R\left( h, \underline{x}(\overline{V}_c[i]) + 1 \right) + \sum_{h=\underline{h}(\overline{V}_c[i]) + 1}^{2^\ell - 1} R\left( h, \underline{x}(\overline{V}_c[i]) \right).$$

We further make the following definitions:

- The probability of rounding the message up to $\underline{x}(\overline{V}_c[i]) + 1$ when $\overline{H}_c[i] = \underline{h}$:

$$\overline{p}_c[i] = \frac{\overline{\mu}_c[i] - R(\overline{H}_c[i], \underline{x}(\overline{V}_c[i]))}{R(\overline{H}_c[i], \underline{x}(\overline{V}_c[i]) + 1) - R(\overline{H}_c[i], \underline{x}(\overline{V}_c[i]))}$$

- The probability of rounding the message down to $\underline{x}(\overline{V}_c[i])$ when $\overline{H}_c[i] = \underline{h}$:

$$\overline{q}_c[i] = 1 - \overline{p}_c[i] = \frac{R(\overline{H}_c[i], \underline{x}(\overline{V}_c[i]) + 1) - \overline{\mu}_c[i]}{R(\overline{H}_c[i], \underline{x}(\overline{V}_c[i]) + 1) - R(\overline{H}_c[i], \underline{x}(\overline{V}_c[i]))}.$$

Then, for any shared randomness value $\overline{H}_c[i] \in \mathcal{H}_\ell$, to-be-quantized value $\overline{V}_c[i] \in [-t_p, t_p]$, and message $x \in \mathcal{X}_b$, the interpolated algorithm works as follows:

$$S(\overline{H}_c[i], \overline{V}_c[i], x) = \begin{cases} 1 & \text{If } \left( x = \underline{x}(\overline{V}_c[i]) \wedge \overline{H}_c[i] > \underline{h} \right) \vee \left( x = \underline{x}(\overline{V}_c[i]) + 1 \wedge \overline{H}_c[i] < \underline{h} \right) \\ \overline{p}_c[i] & \text{If } \left( x = \underline{x}(\overline{V}_c[i]) + 1 \wedge \overline{H}_c[i] = \underline{h} \right) \\ \overline{q}_c[i] & \text{If } \left( x = \underline{x}(\overline{V}_c[i]) \wedge \overline{H}_c[i] = \underline{h} \right) \\ 0 & \text{Otherwise} \end{cases} .$$

$$(11)$$

Namely, if $\overline{H}_c[i] < \underline{h}$, the client deterministically sends $\left(\underline{x}(\overline{V}_c[i]) + 1\right)$ and if $\overline{H}_c[i] > \underline{h}$, the client deterministically sends $\underline{x}(\overline{V}_c[i])$. Finally, if $\overline{H}_c[i] = \underline{h}$, it sends $\left(\underline{x}(\overline{V}_c[i]) + 1\right)$ with probability $\overline{p}_c[i]$ and $\underline{x}(\overline{V}_c[i])$ otherwise. Indeed, by our choice of $\overline{\mu}_c[i]$, the algorithm is guaranteed to be unbiased for all $\overline{V}_c[i] \in [-t_p, t_p]$.

The pseudocode of this variant is given by Algorithm 3.

---

**Algorithm 3** QUIC-FL with client-specific shared randomness and client interpolation

---

**Input:** Bit budget $b$, shared random bits $\ell$, BSQ parameter $p$ and its threshold $t_p$ and precomputed quantiles $\mathcal{A}_{p,m}$, and receiver table $R$. (The table $S$ is not needed.)

---

**Client $c$:**

    1. $\overline{Z}_c \leftarrow \frac{\sqrt{d}}{\|\overline{x}_c\|_2} \cdot T\left(\overline{x}_c\right)$

    2. $\overline{U}_c, \overline{I}_c \leftarrow \left\{\overline{Z}_c[i] \mid \left|\overline{Z}_c[i]\right| > t_p\right\}, \left\{i \mid \left|\overline{Z}_c[i]\right| > t_p\right\}$

    3. $\overline{V}_c \leftarrow \left\{z \in \overline{Z}_c \mid |z| \leq t_p\right\}$

    4. $\overline{H}_c \leftarrow \left\{\forall i : \text{Sample } \overline{H}_c[i] \sim \mathcal{U}[\mathcal{H}_\ell]\right\}$

    5. $\overline{X}_c \leftarrow \left\{\forall i : \text{Sample } \overline{X}_c[i] \sim \left\{x \text{ with prob. } S(\overline{H}_c[i], \overline{V}_c[i], x)\right\}\right\}$       ▷ According to Equation (11)

    6. Send $\left(\|\overline{x}_c\|_2, \overline{X}_c, \overline{U}_c, \overline{I}_c\right)$ to server

---

**Server:**

    7. For all $c$:

    8.      $\overline{H}_c \leftarrow \left\{\forall i : \text{Sample } \overline{H}_c[i] \sim \mathcal{U}[\mathcal{H}_\ell]\right\}$

    9.      $\widehat{\overline{V}}_c \leftarrow \left\{\forall i : R(\overline{H}_c[i], \overline{X}_c[i])\right\}$

    10.      $\widehat{\overline{Z}}_c \leftarrow \text{Merge } \widehat{\overline{V}}_c \text{ and } \left(\overline{U}_c, \overline{I}_c\right)$

    11. $\widehat{\overline{Z}}_{avg} \leftarrow \frac{1}{n} \cdot \sum_{c=0}^{n-1} \frac{\|\overline{x}_c\|_2}{\sqrt{d}} \cdot \widehat{\overline{Z}}_c$

    12. $\widehat{\overline{x}}_{avg} \leftarrow T^{-1}\left(\widehat{\overline{Z}}_{avg}\right)$

---

### F.2 MEMORY REQUIREMENTS

As explained above, the entire algorithm is determined by the server 's table (whose size is $2^{b \cdot \ell}$). The RHT happens in-place, so no additional space is needed other than for holding the gradient. Depending on the implementation, additional memory may be used for (1) a parallel generation of the shared randomness values (2) a parallel computation of the rounding probabilities.

### G PERFORMANCE OF QUIC-FL WITH THE RANDOMIZED HADAMARD TRANSFORM

As described earlier, while ideally we would like to use a fully random rotation on the $d$-dimensional sphere as the first step to our algorithms, this is computationally expensive. Instead, we suggest using a randomized Hadamard transform (RHT), which is computationally more efficient. We formally show below that using RHT has the same asymptotic guarantee as with random rotations, albeit with a larger constant (constant factor increases in the fraction of exactly sent coordinates and *NMSE*). Namely, we show that (1) the expected number of transformed and scaled coordinates that fall outside $[-t_p, t_p]$ (for the same choice of $t_p$ as a function of $p$), is bounded by $3.2p$; (2) that we still get $O(1/n)$ NMSE for any $b \geq 1$. Further, we find that running QUIC-FL with RHT and $b + 1$ bits per quantized coordinate has a lower *NMSE* than QUIC-FL with a uniform random rotation for $p = 2^{-9}$ and any $b \in \{1, 2, 3\}$.

We note that some works suggest using two or three successive randomized Hadamard transforms to obtain something that should be closer to a uniform random rotation Yu et al. (2016); Andoni et al. (2015). This naturally takes more computation time. In our case, and in line with previous works Vargaftik et al. (2021; 2022), we find empirically that one RHT appears to suffice. However,

unlike these works, our algorithm remains **provably unbiased** and maintains the $O(1/n)$ *NMSE* guarantee. Determining better provable bounds using two or more RHTs is left as an open problem.

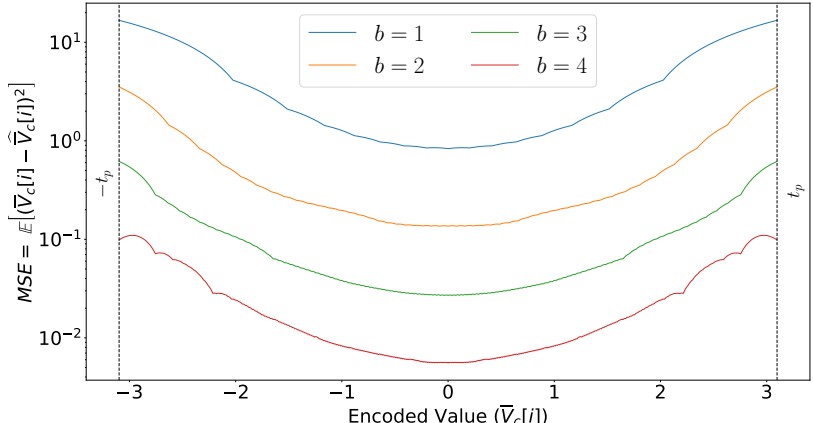

Figure 6: Expected squared error as a function of the encoded value (for $p = \frac{1}{512}, m = 512$).

**Theorem G.1.** *Let $\overline{x} \in \mathbb{R}^d$, let $T_{RHT}(\overline{x})$ be the result of a randomized Hadamard transform on $\overline{x}$, and let $\mathfrak{Z} = \overline{V}_c[i] = \frac{\sqrt{d}}{\|\overline{x}\|_2} T_{RHT}(x)[i]$ be a coordinate in the transformed and scaled vector. For any $p$, $\Pr\left[\mathfrak{Z} \notin [-t_p, t_p]\right] \leq 3.2p$.*

*Proof.* This follows from the theorem by Bentkus & Dzindzalieta (2015) (Theorem G.2), which we restate below. □

**Theorem G.2** (Bentkus & Dzindzalieta (2015)). *Let $\epsilon_1, \ldots, \epsilon_d$ be i.i.d. Radamacher random variables and let $\overline{a} \in \mathbb{R}^d$ such that $\|\overline{a}\|_2^2 \leq 1$. For any $t \in \mathbb{R}$, $\Pr\left[\sum_{i=0}^{d-1} \overline{a}[i] \cdot \epsilon_i \geq t\right] \leq \frac{\Pr[Z \geq t]}{4 \Pr[Z \geq \sqrt{2}]} \approx 3.1787 \Pr[Z \geq t]$, for $Z \sim \mathcal{N}(0, 1)$.*

In what follows, we present a general approach to bound the quantization error of each transformed and scaled coordinate (and thus, the QUIC-FL's *NMSE*). Our method splits $[0, t_p]$ (the argument is symmetric for $[-t_p, 0]$) into several (e.g., three) intervals $\mathfrak{I}_0, \ldots, \mathfrak{I}_w$ (for some $w \in \mathbb{N}^+$), such that the partitioning satisfies two properties:

- The maximal error for the $i$'th interval, $\max_{z \in \mathfrak{I}_i} \mathbb{E}\left[(z - \widehat{z})^2\right]$, is lower than the $j$'th interval, for any $j < i$.

- The probability that a normal random variable $Z \sim \mathcal{N}(0, 1)$ falls outside $\mathfrak{I}_0$ is less than $1/3.2$.

These two properties allow us to use Theorem G.2 to upper bound the resulting quantization error.

We exemplify the method using $p = \frac{1}{512}$, the parameter of choice for our evaluation, although it is applicable to any $p$. Since we believe it provides only a loose bound, we do not optimize the argument beyond showing the technique.

**Theorem G.3.** *Fix $p = \frac{1}{512}$; let $\overline{x}_c \in \mathbb{R}^d$ and denote by $\mathfrak{Z} = \overline{V}_c[i] = \frac{\sqrt{d}}{\|\overline{x}_c\|_2} T_{RHT}(\overline{x}_c)[i]$ its $i$'th coordinate after applying RHT and scaling. Denoting by $E_b = \mathbb{E}\left[(\mathfrak{Z} - \widehat{\mathfrak{Z}_b})^2\right]$ the mean squared error using $b$ bits per quantized coordinate, we have $E_1 \leq 4.831$, $E_2 \leq 0.692$, $E_3 \leq 0.131$, $E_4 \leq 0.0272$.*

*Proof.* We bound the MSE of quantizing $\mathfrak{Z}$, leveraging Theorem G.2. Since the MSE, as a function of $\mathfrak{Z}$, is symmetric around 0 (as illustrated in Figure 6), we analyze the $\mathfrak{Z} \geq 0$ case.

We split $[0, t_p]$ into intervals that satisfy the above properties, e.g., $\mathfrak{I}_0 = [0, 1.5]$, $\mathfrak{I}_1 = (1.5, 2.2]$, $\mathfrak{I}_2 = (2.2, t_p]$. We note that this choice of intervals is not optimized and that a finer-grained partition to more intervals can improve the error bounds. Next, using Theorem G.2, we get that

- $P_0 \triangleq \Pr[\mathfrak{Z} \notin \mathfrak{I}_0] \leq 3.2 \Pr[Z \notin \mathfrak{I}_0] \leq 0.427$.

- $P_1 \triangleq \Pr[\mathfrak{Z} \notin (\mathfrak{I}_0 \cup \mathfrak{I}_1)] \leq 3.2 \Pr[Z \notin (\mathfrak{I}_0 \cup \mathfrak{I}_1)] \leq 0.089$.

Next, we provide the maximal error for each bit budget $b$ and such interval:

|  | $b = 1$ | $b = 2$ | $b = 3$ | $b = 4$ |
|---|---|---|---|---|
| $\mathfrak{I}_0$ | 2.063 | 0.267 | 0.056 | 0.0134 |
| $\mathfrak{I}_1$ | 6.39 | 0.67 | 0.128 | 0.0285 |
| $\mathfrak{I}_2$ | 16.73 | 3.51 | 0.617 | 0.11 |

Table 3: For each interval $\mathfrak{I}_i$, $i \in \{0, 1, 2\}$ and bit budget $b \in \{1, 2, 3, 4\}$, depicted is the maximal MSE, i.e., $\max_{z \in \mathfrak{I}_i} \mathbb{E}\left[(z - \widehat{z})^2\right]$.

Note that for any $b \in \{1, 2, 3, 4\}$, the MSEs in $\mathfrak{I}_2$ are strictly larger than those in $\mathfrak{I}_1$ which are strictly larger than those in $\mathfrak{I}_0$. This allows us to derive formal bounds on the error. For example, for $b = 1$, we have that the error is bounded by

$$E_1 \leq (1 - P_0) \cdot 2.063 + (P_0 - P_1) \cdot 6.39 + P_1 \cdot 16.73 \leq 4.831.$$

Repeating this argument, we also obtain:

$$E_2 \leq (1 - P_0) \cdot 0.267 + (P_0 - P_1) \cdot 0.67 + P_1 \cdot 3.51 \leq 0.692$$
$$E_3 \leq (1 - P_0) \cdot 0.056 + (P_0 - P_1) \cdot 0.128 + P_1 \cdot 0.617 \leq 0.131$$
$$E_4 \leq (1 - P_0) \cdot 0.0134 + (P_0 - P_1) \cdot 0.0285 + P_1 \cdot 0.11 \leq 0.0272. \qquad \square$$

## H EXPERIMENTS WITH ADDITIONAL DISTRIBUTIONS

While QUIC-FL's *NMSE* is largely independent of the input vectors (for a large enough dimension), other algorithms' *NMSE* depends on the inputs. We thus repeat the experiment of Figure 3 for additional distributions, with the results depicted in Figure 7 and Figure 8. As shown, in all cases, QUIC-FL has an *NMSE* that is comparable with that of EDEN.

## I SHAKESPEARE EXPERIMENTS DETAILS

The Shakespeare next-word prediction discussed in §4 was first suggested in McMahan et al. (2017) to naturally simulate a realistic heterogeneous federated learning setting. Its dataset consists of 18,424 lines of text from Shakespeare plays Shakespeare partitioned among the respective 715 speakers (i.e., clients). We train a standard LSTM recurrent model Hochreiter & Schmidhuber (1997) with $\approx 820K$ parameters and follow precisely the setup described in Reddi et al. (2021) for the Adam server optimizer case. We restate the hyperparameters for convenience in Table 4.

| Task | Clients per round | Rounds | Batch size | Client lr | Server lr | Adam's $\epsilon$ |
|---|---|---|---|---|---|---|
| Shakespeare | 10 | 1200 | 4 | 1 | $10^{-2}$ | $10^{-3}$ |

Table 4: Hyperparameters for the Shakespeare next-word prediction experiments.

## J ADDITIONAL EVALUATION

Our code will be released as open source upon publication. As discussed, we use $p = 1/512$, $\ell = 6$ for $b = 1$, $\ell = 5$ for $b = 2$, and $\ell = 4$ for $b \in \{3, 4\}$.

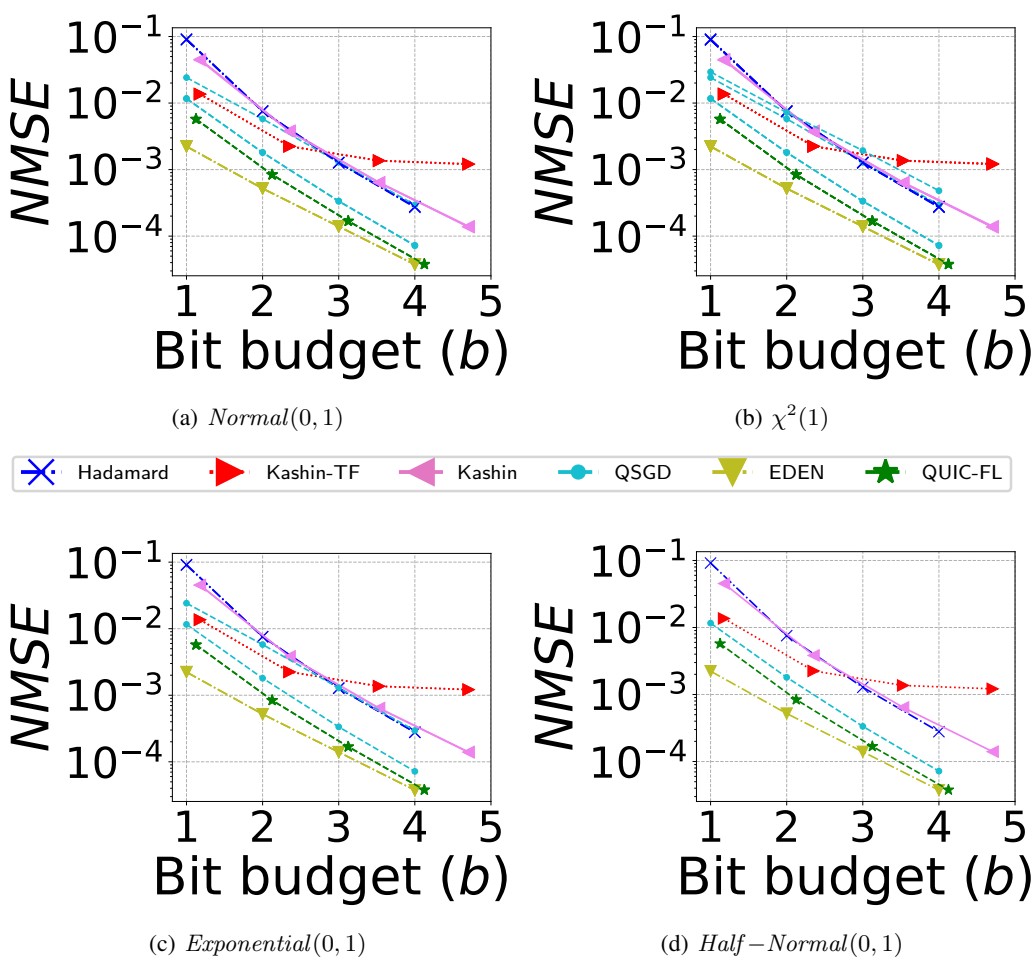

Figure 7: *NMSE* vs. the the bit budget $b$.

## J.1 IMAGE CLASSIFICATION

We evaluate QUIC-FL against other schemes with 10 persistent clients over uniformly distributed CIFAR-10 and CIFAR-100 datasets Krizhevsky et al. (2009). We also evaluate *Count-Sketch* Charikar et al. (2002) (denoted CS), often used for federated compression schemes (e.g., Ivkin et al. (2019)) and EF21 Richtárik et al. (2021) a recent SOTA error-feedback framework that uses top-k as a building block with $k = 0.05 \cdot d$ (translates to 1.6 bits per-coordinate ignoring the overhead of indices encoding overhead). For QSGD, we use twice the bandwidth of the other algorithms (one bit for sign and another for stochastic quantization). We note that QSGD also has a more accurate variant that uses variable-length encoding Alistarh et al. (2017). However, it is not GPU-friendly, and therefore, as with other variable-length encoding schemes, as we have discussed previously, we do not include it in the experiment.

For CIFAR-10 and CIFAR-100, we use the ResNet-9 He et al. (2016) and ResNet-18 He et al. (2016) architectires, and use learning rates of 0.1 and 0.05, respectively. For both datasets, the clients perform a single optimization step at each round. Our setting includes an SGD optimizer with a cross-entropy loss criterion, a batch size of 128, and a bit budget $b = 1$ for the DME methods (except for EF21 and QSGD as stated above). The results are shown in Figure 9, with a rolling mean average window of 500 rounds. As shown, QUIC-FL is competitive with EDEN and the Float32 baseline and is more accurate than other methods.

Next, we repeat the above CIFAR-10 and CIFAR-100 experiments with the same bandwidth budgets but consider a cross-device setup with the following changes: there are 50 clients (instead of 10) and

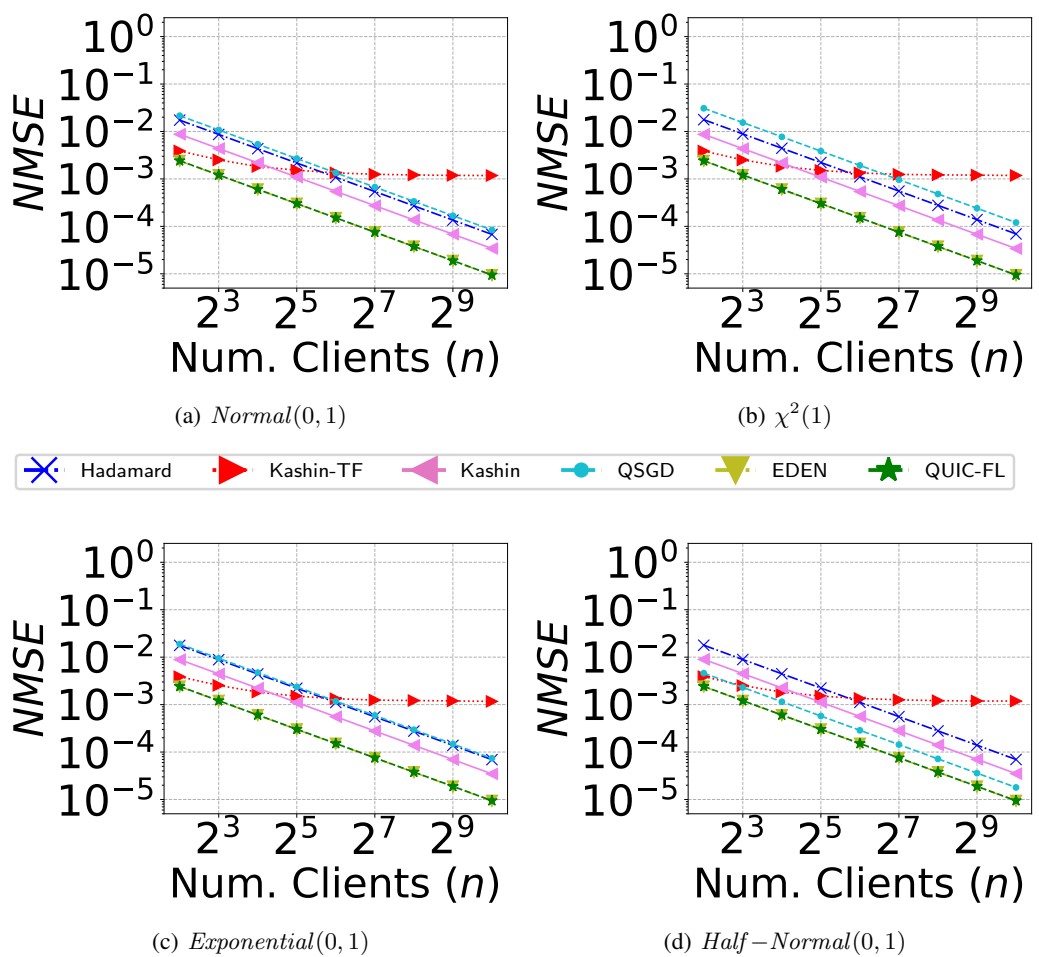

Figure 8: *NMSE* vs. the the number of clients $n$.

at each training round, 10 out of 50 clients are randomly selected and perform training over 5 local steps (instead of 1).

Figure 10 shows the results with a rolling mean window of 200 rounds. Again, QUIC-FL is competitive with the asymptotically slower EDEN and the uncompressed baseline. Kashin-TF is less accurate, followed by Hadamard.

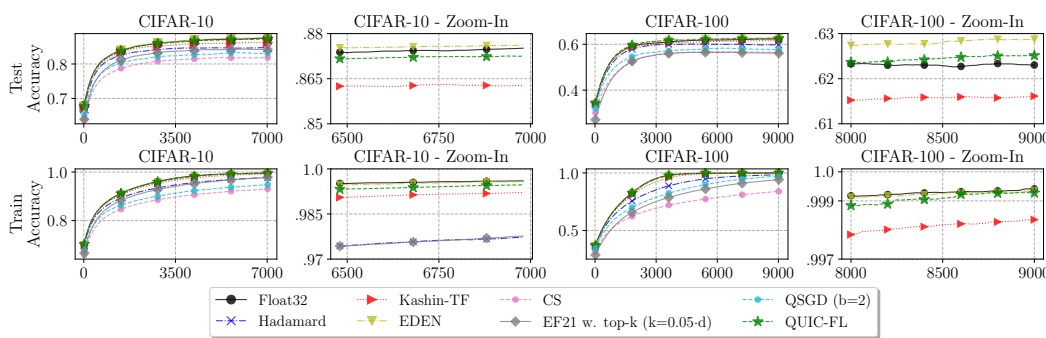

Figure 9: Cross-silo federated learning.

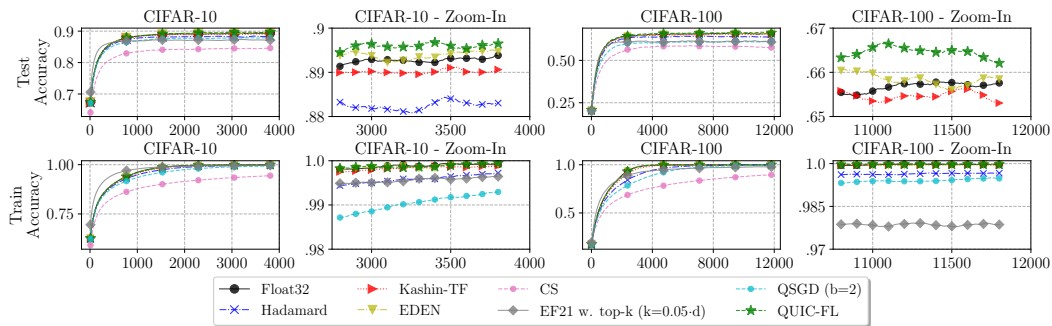

Figure 10: Cross-device federated learning.

## J.2 DME AS A BUILDING BLOCK

We pick EF21 Richtárik et al. (2021) as an example framework that uses DME as a building block. In the paper, EF21 is used in conjunction with top-$k$ as the compressor that is used by the clients to transmit their messages, and the mean of the messages is estimated at the server. As shown in Figure 11, using EF21 with QUIC-FL instead of top-$k$ significantly improves the accuracy of EF21 despite using less bandwidth. For example, top-$k$ with $k = 0.1 \cdot d$ needs to use 3.2 bits per coordinate on average to send the values (in addition to the overhead of encoding the indices) while having accuracy that is lower than EF21 with QUIC-FL and $b = 2$ bits per coordinate.

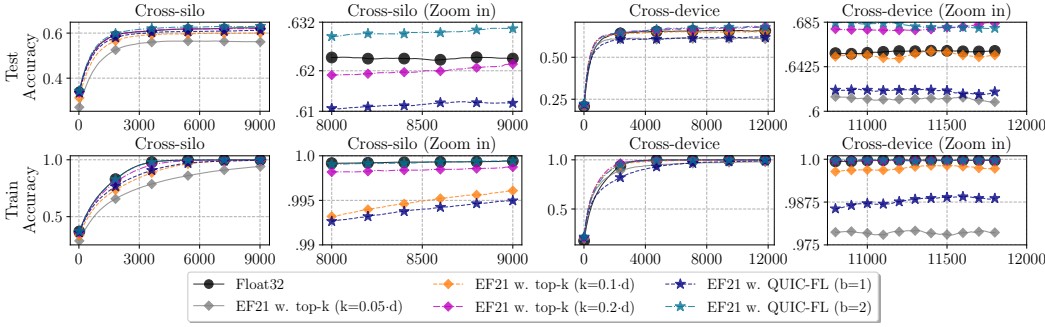

Figure 11: The accuracy of EF21 with top-$k$ and QUIC-FL as building blocks for DME.

## J.3 DISTRIBUTED POWER ITERATION

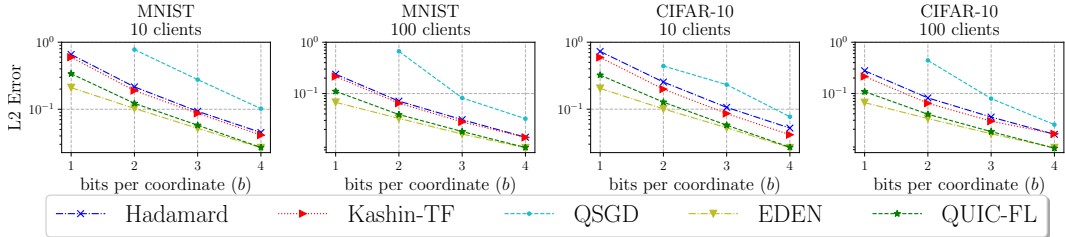

Figure 12: Distributed power iteration of MNIST and CIFAR-10 with 10 and 100 clients.

We simulate 10 clients that distributively compute the top eigenvector in a matrix (i.e., the matrix rows are distributed among the clients). Particularly, each client executes a power iteration, compresses its

top eigenvector, and sends it to the server. The server updates the next estimated eigenvector by the averaged diffs (of each client to the eigenvector from the previous round) and scales it by a learning rate of $0.1$. Then, the estimated eigenvector is sent by the server to the clients and the next round can begin.

Figure 12 presents the L2 error of the obtained eigenvector by each compression scheme when compared to the eigenvector that is achieved without compression. The results cover bit budget $b$ from one bit to four bits for both MNIST and CIFAR-10 Krizhevsky et al. (2009); LeCun et al. (1998; 2010) datasets. Each distributed power iteration simulation is executed for 50 rounds for the MNIST dataset and for 200 rounds for the CIFAR-10 dataset.

As shown, QUIC-FL has an accuracy that is competitive with that of EDEN (especially for $b \geq 2$) while having asymptotically faster decoding, as EDEN requires decompressing the vector for each client independently. At the same time, QUIC-FL is considerably better in terms of accuracy than other algorithms that offer fast decoding time. Also, Kashin-TF is not unbiased (as illustrated by Figure 2), and is, therefore, less competitive for a larger number of clients.

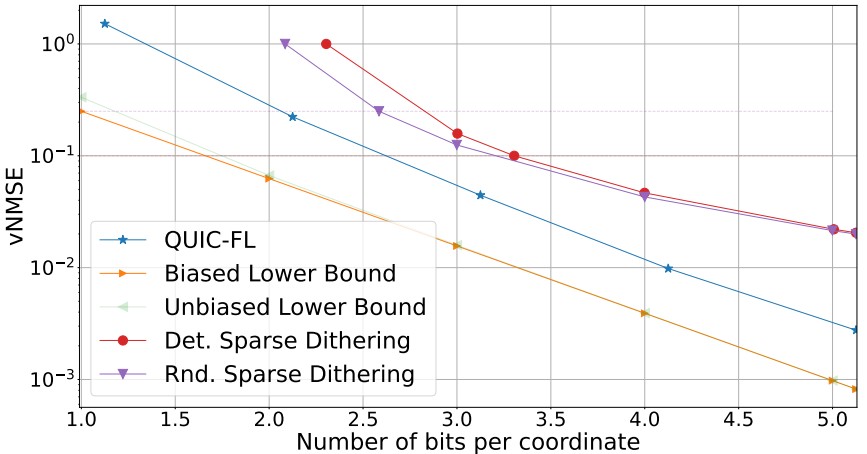

Figure 13: Comparison with Sparse Dithering.

## J.4  COMPARISON WITH SPARSE DITHERING

We compare QUIC-FL with Sparse Dithering (SD) Albasyoni et al. (2020). As shown in Figure 13, QUIC-FL is markedly more accurate for the range of bit budgets ($b \in \{1, 2, 3, 4, 5\}$) that it supports. The figure includes both the deterministic and randomized versions of SD.

The markers mark the evaluated points. QUIC-FL is configured with $p = 2^{-9}$, and thus its per-coordinate bandwidth is non-integer to factor in the coordinates sent exactly.

Further, our algorithm is proven to be GPU friendly, while we cannot determine whether the components of the Sparse Dithering algorithm can be efficiently implemented. The paper does not include a runtime evaluation that we can compare with.

