# OpenReview forum: "Accelerating Federated Learning with Quick Distributed Mean Estimation"
_ICLR.cc/2024/Conference — Submitted to ICLR 2024_

### Official Review · Reviewer_fdHV · 2023-10-22

**Soundness:** 2 fair
**Presentation:** 1 poor
**Contribution:** 3 good
**Rating:** 5
**Confidence:** 4

**Summary:**

This works addresses the problem of communication-efficient distributed mean estimation (DME), a common subroutine in distributed optimization, and improves the SOTA of quantization techniques. It proposes QUIC-FL, the first quantization scheme that is efficient in both encoding and decoding and achieves the optimal NMSE, the metric of the estimation error, of $O(1/n)$ at the same time, where $n$ is the number of clients. QUIC-FL is based on two key ideas: 1) Bounded support quantization (BSQ), which sends a few large coordinates exactly and quantizes only the rest few. 2) An optimization framework that optimizes the set of quantization values to further reduce the estimation error, based on the limiting distribution of transformed coordinates of the client vectors, i.e., the normal distribution. Furthermore, QUIC-FL discusses the usage of client-specific shared randomness and the RHT rotation to practically gain constant improvement in the estimation error. Finally, extensive experiments show advantages of QUIC-FL compared to several SOTA quantization schemes in terms of encoding time, decoding time and NMSE.

**Strengths:**

In terms of originality, this paper combines several existing approaches and improve the SOTA of quantization schemes.

In terms of quality, the paper analyzes and empirically shows an improved performance of the proposed QUIC-FL in terms of computational efficiency and the estimation error, against several SOTA baselines.

In terms of clarity, the paper conveys the key ideas in the design of QUIC-FL.

In terms of significance, the problem of communication-efficient DME the paper addresses is important. Improvement in quantization schemes is always welcomed.

**Weaknesses:**

The presentation of this draft needs to be greatly improved.

-	The abstract is not informative at all. The reader has no idea about the techniques this work uses to improve DME and the novelty of the techniques after reading it. It should at least mention, for example, one idea QUIC-FL builds on is BSQ which sends exactly a few large coordinates and quantizes the rest small ones.

-	In Introduction, the paragraph starting with “For example, in Suresh et al. 2017 …”, it is mentioned the entropy encoding is “compute-intensive”. How long does the decoding time of this approach take? Table 1 does not include entropy encoding as a baseline.

-	It is mentioned in Introduction that the decoding procedure of “entropy encoding” from a previous work is “compute-intensive”. What is the time complexity of this decoding procedure? It seems this approach is not included in Table 1 as a baseline for comparison.

-	The Lloyd-Max quantizer appears at several places in the work, e.g., in Introduction, and in serving as the Lower Bound in Section 3.5. However, this work does not introduce this quantizer properly. Can the authors give a brief introduction of this quantizer and in which cases is it optimal?

-	The paragraph “while the above methods suggest …” is abrupt and confusing. It’d be better to move this paragraph before surveying existing quantization techniques in Introduction.

-	In Section 2 preliminaries, it would be clearer if “unbiased algorithms and independent estimators” can be formally and clearly defined. Minor issue: “we that NMSE …” => “we want that NMSE …”

-	The uniform random rotation appears at several places in the work. What exactly is this rotation? Is it a uniform Gaussian random rotation?

-	In Section 3.5 “accelerating QUIC-FL with RHT”, “adversarial vectors” are mentioned but not introduced. It is confusing how the proposed approach compares against DRIVE and EDEN in terms of the application to “adversarial vectors”.

-	In Section 3.5 “accelerating QUIC-FL with RHT”, it states “the result does not have the additive NMSE term is [because] we directly analyze the error for the Hadamard-rotated coordinates”. Can the author be more formal and specific how the analysis here is different from the one in Theorem 3.1?

-	Several places in the work states in plain text the performance of QUIC-FL with different values of its hyperparameters. For example, in Section 3.5 “accelerating QUIC-FL with RHT”, it states the NMSE of QUIC-FL is bounded by … with $b = 1,2,3,4$. It’d be easier for the readers if those numbers can be turned into figures.

This work claims the proposed QUIC-FL has NMSE $O(1/n)$.  However, by Theorem 3.1, the actual NMSE is indeed $O(1/n \cdot \sqrt{\log d / d})$. It does not seem to be OK to omit the $O(\sqrt{\log d / d})$ term and directly write $O(1/n \cdot \sqrt{\log d / d}) = O(1/n)$.

**Questions:**

$\textbf{Optimizing the quantization values. }$
In Section 3.3 “distribution-aware unbiased quantization”, this work proposes two optimization problems to find the optimal quantization values to reduce NMSE. In the first optimization problem on page 4, the notations $S(z, x)$ and $R(x)$ are a bit confusing. Are $S$ and $R$ two functions to be optimized? Is $R(x)$ essentially a vector of $2^b$ variables? Is $S(z, x)$ a continuous function?

Similarly, in the second optimization problem (i.e., the discretized version of the first problem), is $S’(i, x)$ essentially a vector of $m \cdot 2^b$ variables to be optimized?

Since the number of variables to be optimized is on the order of $2^b$, how efficient is the second optimization problem?

$\textbf{Communication cost. }$
One concern is that the proposed QUIC-FL requires extra bits to send a few large coordinates exactly along with their indices, while the baseline quantization schemes usually allocate a fixed number of bits per coordinate. This makes QUIC-FL use more communication cost compared to the baseline. And hence it might not be fair to directly compare QUIC-FL’s NMSE against that of the baselines. How does the author compare in Table 1? Also, how does the author address this in the experiments?

$\textbf{Optimality. }$
It is mentioned at several places that the optimal NMSE of any quantization is $O(1/n)$ (this lower bound is in terms of the number of clients only, I presume). This is not rigorous in the draft. Can the authors cite the theorems that indicates the optimality?

The draft claims QUIC-FL achieves a “near-optimal” NMSE. However, it seems this lower bound is only empirically obtained using the Lloyd-Max quantizer in Section 3.5. “Near-optimal” specifically means one can theoretically show the algorithm achieving optimality (e.g., close to a lower bound), up to a logarithmic factor. And so it might not be appropriate to claim the “near-optimality” of QUIC-FL.

---

> ### Author Response · Authors · 2023-11-15
> **Author's response**
>
> Thank you for the review.
>
> ---
>
> *Weaknesses*
>
> ---
>
> **The abstract is not informative at all. The reader has no idea about the techniques this work uses to improve DME and the novelty of the techniques after reading it. It should at least mention, for example, one idea QUIC-FL builds on is BSQ which sends exactly a few large coordinates and quantizes the rest small ones.**
>
> We will revisit the abstract to make it more informative.
>
> **In Introduction, the paragraph starting with “For example, in Suresh et al. 2017 …”, it is mentioned the entropy encoding is “compute-intensive”. How long does the decoding time of this approach take? Table 1 does not include entropy encoding as a baseline.**
>
> **It is mentioned in Introduction that the decoding procedure of “entropy encoding” from a previous work is “compute-intensive”. What is the time complexity of this decoding procedure? It seems this approach is not included in Table 1 as a baseline for comparison.**
>
> In theory, entropy encoding requires $O(d)$ time. However, this operation is not GPU-friendly and is thus much slower in practice.
> In the paper “DoCoFL: Downlink Compression for Cross-Device Federated Learning” (ICML 2023), the authors performed timing measurements of entropy encoding schemes and found it to be ~ two orders of magnitude slower compared to EDEN, which runs on a GPU (QUIC-FL also runs on a GPU and is faster than EDEN as we show in the paper).
>
> The Lloyd-Max quantizer is a biased quantizer that yields good quantization levels for a given distribution. In some cases, the outcome may be suboptimal (see [A], Example 3.2.1 ). However, the quantizer is known to be optimal for the log-concave distributions (see, e.g., [B]), including the Guassian, which is the focus of our work.
>
> [A] MIT lecture notes https://ocw.mit.edu/courses/6-450-principles-of-digital-communications-i-fall-2006/926689aaa62a0315473fa9b982de1b07_book_3.pdf
>
> [B] Huang, Bormin, and Jing Ma. "On asymptotic solutions of the Lloyd-Max scalar quantization." 2007 6th International Conference on Information, Communications & Signal Processing. IEEE, 2007.
> https://ieeexplore.ieee.org/document/4449824.
>
> **The paragraph “while the above methods suggest …” is abrupt and confusing. It’d be better to move this paragraph before surveying existing quantization techniques in Introduction.**
>
> **In Section 2 preliminaries, it would be clearer if “unbiased algorithms and independent estimators” can be formally and clearly defined. Minor issue: “we that NMSE …” => “we want that NMSE …”**
>
> We will revise these accordingly.
>
> **The uniform random rotation appears at several places in the work. What exactly is this rotation? Is it a uniform Gaussian random rotation?**
>
> Random Matrix Theory (RMT) was introduced by Wishart [C] in 1928, and includes rotating the vector by multiplying it with a “rotation matrix”, which is a random orthogonal matrix (see e.g., [D] for more details and methods for its generation. We will discuss this in the paper.
>
> [C] Wishart J. The generalised product moment distribution in samples from a normal multivariate population. Biometrika 20A, 32–52 (1928).
> [D] Martin A Tanner and Ronald A Thisted. Generation of Random Orthogonal Matrices. Journal of the Royal Statistical Society. Series C (Applied Statistics), 31(2):190–192, 1982.
>
> **In Section 3.5 “accelerating QUIC-FL with RHT”, “adversarial vectors” are mentioned but not introduced. It is confusing how the proposed approach compares against DRIVE and EDEN in terms of the application to “adversarial vectors”.**
>
> We meant that there are inputs for which DRIVE and EDEN fail to give an $O(1/n)$ NMSE when using RHT, as we explain in Appendix A. This is not a distinct application but rather an acknowledgment that DRIVE and EDEN are not suitable for worst-case situations.
>
> **In Section 3.5 “accelerating QUIC-FL with RHT”, it states “the result does not have the additive NMSE term is [because] we directly analyze the error for the Hadamard-rotated coordinates”. Can the author be more formal and specific how the analysis here is different from the one in Theorem 3.1?**
>
> In Theorem 3.1., we analyze the error that arises from approximating the distribution after a uniform random rotation and show that it is manifested in the $O(\sqrt{\log d / d})$ additive error term. The analysis in Section 3.5 directly analyzes the distribution that results from applying RHT, and thus no additional error term is needed.
>
> **Several places in the work states in plain text the performance of QUIC-FL with different values of its hyperparameters. For example, in Section 3.5 “accelerating QUIC-FL with RHT”, it states the NMSE of QUIC-FL is bounded by … with $b=1,2,3,4$. It’d be easier for the readers if those numbers can be turned into figures.**
>
> We will add the figures to the paper.

---

> > ### Author Response · Authors · 2023-11-15
> > **Author's response (cont')**
> >
> > ---
> >
> > *Weaknesses (cont')*
> >
> > ---
> >
> > **This work claims the proposed QUIC-FL has NMSE $O(1/n)$. However, by Theorem 3.1, the actual NMSE is indeed $O(1/n\cdot \sqrt{\log d/d})$. It does not seem to be OK to omit the $O(\sqrt{\log d/d})$ term and directly write $O(1/n\cdot \sqrt{\log d/d}) = O(1/n)$.**
> >
> > Note that by Theorem 3.1, the NMSE is  $$NMSE = \frac{1}{n} \cdot \mathbb E[(Z {-} \widehat{Z})^2 ]+ O(\frac{1}{n} \cdot\sqrt{\frac{\log d}{d}}).$$
> >
> > We have that $\mathbb E[(Z {-} \widehat{Z})^2 ] = \Theta(1)$ (e.g., $\mathbb E[(Z {-} \widehat{Z})^2 ]=0.00982$ with $b=4$). Therefore, the error is $NMSE = \Theta(\frac{1}{n}) + O(\frac{1}{n} \cdot\sqrt{\frac{\log d}{d}}) = \Theta(\frac{1}{n})$ as $\sqrt{\frac{\log d}{d}} = o(1)$ (recall that $d$ is the dimension of the vector and thus $\frac{\log d}{d} < 1$ for any $d$).
> >
> > ---
> >
> > *Questions*
> >
> > ---
> >
> > **Optimizing the quantization values**
> >
> > Indeed, as shown in the program, the goal is to optimize the error as a function of $S,R$. $R$ is indeed a set of $2^b$ variables (or $2^{b+\ell}$ after the extension to shared randomness in section 3.5) and $S$ can be any probability distribution over the messages in $X$ given an input $z$. Note that we later discretize the problem (pages 5-7), effectively turning $S$ into a discrete set of variables (indeed, $m\cdot 2^b$ for the problem on page 5, increasing to $m\cdot 2^{b+\ell}$ for the problem with shared randomness in page 7).
> >
> > The complexity of the solver that we use is therefore exponential in $b$ and $\ell$; however, we do not need to run it when encoding or decoding! We ran the solver once for any $b,\ell$ combination offline, allowing us to use its solution on any dimension and input vector.
> >
> > **Communication cost**
> >
> > To make a fair comparison, note that we allow the baselines to send more than $b$ bits per coordinate. As QUIC’s default setting uses $p=1/512$, and we can send each exact coordinate using $64$ bit (in a simple encoding that uses $32$ bits for the value and another $32$ bits for the index), we consider QUIC-FL to be using $b+64/512=b+1/8$ bits per coordinate, e.g., in the leftmost subfigure of Figure 3, which shows the exact bit overhead for each algorithm.
> >
> > Also, note that some of the baselines in Figure 4 do not use exactly $b$ bits for coordinate either; e.g., Kashin’s representation requires at least $1.3\cdot b$ bits as it projects the vector into dimension $\lambda\cdot d$ before applying the quantization. Similarly, we did not “charge” QSGD for the sign bit that it sends in addition to each quantized coordinate (i.e., it effectively runs with $b+1$ bits in Figure 4).
> >
> > **Optimality**
> >
> > The optimality arises from a $\Omega(1)$ information-theoretic lower bound on vNMSE of any quantization; see, e.g., [E]. We will clarify this in the paper.
> >
> > [E] Mher Safaryan, Egor Shulgin, and Peter Richtárik. Uncertainty principle for communication compression in distributed and federated learning and the search for an optimal compressor. Information and Inference: A Journal of the IMA, 2020.
> >
> > Regarding the near-optimality claim: QUIC-FL achieves $O(1/n)$ NMSE using $b= O(1)$, which is asymptotically optimal. We use the term “near-optimal” as the constant is not tight. We also note that Lloyd-Max is known to be optimal for log-concave distributions (including Gaussian), so we get a rigorous bound on the constant gap between the optimum and our solution.

---

### Official Review · Reviewer_umvY · 2023-10-31

**Soundness:** 4 excellent
**Presentation:** 3 good
**Contribution:** 3 good
**Rating:** 6
**Confidence:** 2

**Summary:**

This paper introduces a novel quantization algorithm with application in federated learning. The key parameters are determined via a constraint optimization problem. Notably, coordinates above the determined threshold are explicitly transmitted to the server, while other values are quantized. To simplify decoding on the server, the clients apply a common preprocessing rotation to the local vectors. This step also modifies the distribution of the coordinates to reduce quantization errors.

**Strengths:**

+ The overview of the state-of-the-art is well presented, with comparisons with existing methods.

+ The authors demonstrate that the encoding and decoding times of their method are comparable to those of competitors, but with greater precision.

**Weaknesses:**

* The authors propose interesting contributions, although some ideas have similarities with existing work.

**Questions:**

* The parameters of the quantization set $\mathcal{Q}_{b,p}$ are obtained by solving a problem that considers the quantiles of a truncated Gaussian distribution. In practice, do the entries of the rotated vectors follow this Gaussian distribution? Instead of considering the standard normal distribution $\mathcal{N}(0,1)$, would it be possible to approximate the coordinate distribution by a parametric distribution?

* What is the complexity of the optimization problem for determining $b, p, t_p$?

**Requested Changes:**

* Page 3: "have" is probably missing between "we" and "that" at the end of the paragraph "Problems and Metric".

* Page 5: in the definition of $\mathcal{A}\_{p,m}$, I think $A\_{p,m}(i)$ should be replaced by $\mathcal{A}_{p,m}(i)$.

* Page 7: "the sender sends the message $x$", perhaps should be replaced by "the sender sends the message $R(h,x)$".

**Details Of Ethics Concerns:**

I don't see any major concerns, the paper designs a novel quantization method with a particular interest in federated learning.

---

> ### Author Response · Authors · 2023-11-15
> **Author's response**
>
> Thank you for the review.
>
> Regarding the first question: Yes! One can use our approach to optimize the quantization for the actual distribution (shifted-Beta, as we explain in the paper - page 6). We knowingly chose to optimize it for the normal distribution instead as (1) it is independent of the dimension of the input vectors, (2) it is a good approximation and the error that arises from it diminishes very quickly with the dimension.
> This way, we can run the solver once and get a solution that is applicable across a range of vector dimensions and learning tasks. By restricting the number of solver invocations, we can also run the solver to yield a better approximation (e.g., more quantiles and shared randomness values) for the same overall computation/time.
>
> The runtime of the solver is exponential in $b$ (and is largely independent of $p$ and $t_p$), but we only need to run it once (offline) for each value of $b$, and the values of interest are small (e.g., $b\in\{1,2,3,4\}$).
>
> We will clarify these in the paper.
>
> Regarding the requested changes:
> We will fix the first two. Thank you!
> With respect to the third, it should indeed be $x$ and not $R(h,x)$, which is the value that the receiver estimates the coordinate at given the shared randomness value $h$ and message $x$.

---

### Official Review · Reviewer_oj6v · 2023-11-03

**Soundness:** 3 good
**Presentation:** 3 good
**Contribution:** 3 good
**Rating:** 6
**Confidence:** 3

**Summary:**

In this work, the authors propose a distributed mean estimator, namely QUIC-FL, to estimate the mean of n vectors in a distributed setting. Their method achieves the optimal $O(\frac{1}{n})$ NMSE (normalized mean squared error). They provide asymptotic improvement to either encoding complexity or decoding complexity (or both) with respect to the existing methods providing $O(\frac{1}{n})$ NMSE guarantees.

**Strengths:**

I found the introduction of bounded support quantization and its use to achieve $O(\frac{1}{n})$ NMSE interesting. I generally liked the presentation and clarity of the paper. The claims have been repeated at times, I believe for emphasis, but otherwise, it is a well-written paper. I also liked the way authors have placed their work with respect to the existing works. They have also provided a good set of numerical experiments to validate their theory.

**Weaknesses:**

The gain in accuracy seems marginal (if any) as compared to EDEN empirically. The proposed method does perform better in terms of decoding time, but decoding time is usually not a big concern when it is done in a centralized server with sufficient processing power.

**Questions:**

I am slightly confused by the following statement on page 4.

"Empirically, sending the indices using ...as $p . \log d << 1$ in our settings, resulting in fast processing time and small bandwidth overhead."

Does this mean that $p$ is not kept constant? If that is the case, then shouldn't NMSE have an order of $O(\frac{\log d}{n})$?

---

> ### Author Response · Authors · 2023-11-15
> **Author's response**
>
> Thank you for the review.
>
> Indeed, we do not claim that QUIC-FL is generally more accurate than EDEN in practice. Our contribution is in getting a comparable accuracy with asymptotically faster decoding time.
>
> Moreover, our contribution is not just in improving the decoding complexity, but also in providing stronger **worst-case guarantees**. Namely, EDEN has an NMSE of $O(1/n)$ only when using uniform random rotation (which requires $O(d^3)$ time and is computationally intensive) and $O(1)$ when using RHT (as the quantization is biased; see Appendix A for details). In contrast, QUIC-FL has $O(1/n)$ NMSE even when using RHT.
>
> Regarding the decoding time, as we convey in the paper:
> According to recent sources (e.g., see the “Federated Learning with Formal Differential Privacy Guarantees” blog post by Google), cross-device FL training procedures can accommodate many thousands of devices per round. Current works (see “Federated Learning with Formal Differential Privacy Guarantees,” SysML 2019) even discuss using tens of thousands of clients per round. In such a setup, if one wishes to use the most accurate available DME techniques, asymptotically faster aggregation time may translate to significant savings in time and/or resources, even if the aggregation is performed on a powerful server.
>
> In light of our consistent findings and the existing evidence that suggests such savings translate to reduced latency (“..devices have significantly higher-latency, lower-throughput connections..”, Federated learning: Collaborative machine learning without centralized training data, Google blog), we believe our approach will indeed lead to improved real-world deployment.
>
> Regarding the overhead of encoding the indices – we do consider $p$ to be a constant (e.g., 1/512). You are correct that if we encode the indices explicitly, this will cost $O(d\log d\cdot p)$ bits ($O(d\log d)$ if $p$ is constant). In theory (and in practice), we can simply use $d$ bits to encode the indices in a 0-1 vector for $d$-bit overhead for all indices combined in all cases. In practice though, as $p$ is considerably smaller than $1/\log d$, it is cheaper to send the indices, and this was the point of our statement.
>
> In summary, the theoretical results can be achieved by sending a bit vector, and thus we achieve the $O(1/n)$ bound with $b=O(1)$. In practice, for reasonable $p$, $d$, explicit encoding of the $p$-fraction of the coordinates is cheaper.
>
> We will clarify these points in the paper.

---

### Official Review · Reviewer_13gx · 2023-11-04

**Soundness:** 3 good
**Presentation:** 3 good
**Contribution:** 3 good
**Rating:** 5
**Confidence:** 5

**Summary:**

The authors study Distributed Mean Estimation problem (DME) where $n$ clients communicate a representation of a $d$-dimensional vector to a parameter server which estimates the vectors’ mean.

I think the overall presentation of the paper, for example providing Figure 1 is quite helpful to help readers understand the main contribution of this paper. To the best of my knowledge, related work have been covered sufficiently.

In terms of technical contribution, this paper is built on the previous literature DRIVE and EDEN and improves Encoding and Decoding complexity bounds under similar normalized mean squared error bounds. It seems the main advantage of QUIC-FL comes from the tailored random rotation preprocessing which reduces the constant in the NMSE error bound for small values of $p$.

I have an overall positive impression about this work, while I think there are rooms for improvement that will be discussed in the following.

The authors provide PuTorch and TensorFlow implementation and show improvements over QSGD Hadamard, and Kashin. The improvements over DRIVE and EDEN is somehow marginal. It will be also very helpful for the readers if the authors elaborate on the discussion after Theorem 3.1.

**Strengths:**

I think the paper is overall quite well-written.

The related work is comprehensive. I also think it is very nice that that the authors show transparently the superiority of EDEN on low bit-width region in terms of NMSE.

I also like the overall flow of the paper including the intuition provided by authors within the algorithmic description.

**Weaknesses:**

The authors provide PuTorch and TensorFlow implementation and show improvements over QSGD Hadamard, and Kashin.
The improvements over DRIVE and EDEN is somehow marginal.

I appreciate that the authors show the superiority of EDEN on low bitwidth region in terms of NMSE.

I was just wondering whether the authors can come up with a hybrid type method that enjoys the NMSE of EDEN while have similar coding time improvements of QUIC-FL?

------------------

I appreciate the discussion after Theorem 3.1 regarding $\mathrm{E}\big[\big(Z-\hat Z\big)^2\big]$. However, it will be still great if the authors provide an explicit error bounds in terms of $b,p,d$. In the current form, it is a bit difficult to provably show the theoretical improvement.

**Questions:**

I was just wondering whether the authors can come up with a hybrid type method that enjoys the NMSE of EDEN while have similar coding time improvements of QUIC-FL?


Could the authors provide a more explicit bound in Theorem 3.1?

I will be willing to increase my scores during the rebuttal period.

---

> ### Author Response · Authors · 2023-11-15
> **Author's response**
>
> Thank you for the review.
>
> Unfortunately, QUIC-FL’s faster decoding relies on the quantization itself being unbiased, while EDEN uses an optimized biased quantization of the transformed coordinates.  EDEN becomes unbiased for a single sender only after the inverse rotation and scaling are applied; however, this also requires users to use different rotations because otherwise, the senders’ results are not independent. EDEN’s better vNMSE is a direct implication of biased quantizations being capable of lower error rates than unbiased ones. We do not see an approach that allows QUIC-FL to apply different rotations to different senders and maintain its decoding time. In particular, because QUIC-FL uses the same rotation for all senders, the unbiasedness comes from a randomized rounding using the same rotation (unlike EDEN’s deterministic rounding, with different random rotations). We do not see how a hybrid can be achieved.
>
> Regarding 3.1, the term $\mathbb E[(Z-\widehat Z)^2]$ is independent of $d$ and only depends on $b,p$. We present it in this way to allow one to understand how further improvements in the unbiased quantization of normal random variables (e.g., if one has a more powerful solver or is able to solve the continuous variant) translate directly to improvements in the vNMSE bound. We can provide explicit tables that give bounds for meaningful values of $b,p$. One such result is given in Thm G.3 for running QUIC-FL with RHT. Here, we also give the vNMSE results for QUIC-FL with a uniform random rotation, which are closer to the error in practice even when using RHT:
>
> p = 1/128:
>
> b = 1, $\mathbb E[(Z-\widehat Z)^2]$ = 1.408
> b = 2, $\mathbb E[(Z-\widehat Z)^2]$ = 0.201
> b = 3, $\mathbb E[(Z-\widehat Z)^2]$ = 0.0398
> b = 4, $\mathbb E[(Z-\widehat Z)^2]$ = 0.00859
>
> p=1/512:
>
> b = 1, $\mathbb E[(Z-\widehat Z)^2]$  = 1.517
> b = 2, $\mathbb E[(Z-\widehat Z)^2]$ = 0.223
> b = 3, $\mathbb E[(Z-\widehat Z)^2]$ = 0.0444
> b = 4, $\mathbb E[(Z-\widehat Z)^2]$ = 0.00982
>
> In order to get a general formula for uniform random rotations, we can consider uniform quantization (we do at least as good as this). The bound that depends on $b,T_p$ would be  $O(T_p^2 \cdot 4^{-b})$.
> Recall that for $p>0$ we have $T_p = \Phi^{-1}(1-p/2)$, which gives a bound in terms of $p$.
>
> We will add this discussion to the paper.

---

> > ### Comment · Reviewer_13gx · 2023-11-22
> > **After Rebuttal**
> >
> > I would like to thank the authors for their response. I went through Theorem G.3 and noticed that your upper bound is indeed "per coordinate" so the final MSE is in $O(d)$.
> >
> > On Table 1, the authors claim that the NMSE for QSGD is $O(d/n)$, which is wrong. It is indeed $O(\sqrt{d})$ (their $n$ in their Lemma 3.1 is your $d$). For this reason and also the comment raised by Reviewer fdHV regarding the claims in the paper, unfortunately, I do not recommend acceptance.

---

> > > ### Author Response · Authors · 2023-11-22
> > > **Response**
> > >
> > > We thank the reviewer for the further comment. Please see below.
> > >
> > > *I went through Theorem G.3 and noticed that your upper bound is indeed "per coordinate" so the final MSE is in O(d).*
> > >
> > > There seems to be a slight confusion here. The quantization error for a normal(0,1) random variable is indeed constant, as we prove in G.3.
> > > However, this applies to the **scaled** coordinates; if all coordinates were normal(0,1), the norm of the vector would be $O(d)$, so while the MSE is indeed $O(d)$, the vNMSE would be $O(1)$, and thus the NMSE $O(1/n)$, as stated.
> > >
> > > *On Table 1, the authors claim that the NMSE for QSGD is $O(d/n)$, which is wrong.
> > > It is indeed $O(\sqrt d)$ (their $n$ in their Lemma 3.1 is your $d$).*
> > >
> > > Thanks for pointing it out! The vNMSE for QSGD is indeed $O(\sqrt d)$, and thus its NMSE is $O(\sqrt d / n)$.
> > > Notice, however, that this is still not competitive with the other algorithms.
> > >
> > > *For this reason and also the comment raised by Reviewer fdHV regarding the claims in the paper, unfortunately, I do not recommend acceptance.*
> > >
> > > Please note that we provided individual responses to each of Reviewer fdHV's comments. If any of these is not factual or does not answer the concern, please let us know, and we can clarify it further.

---

### Author Response · Authors · 2023-11-15
**Rebuttal**

We thank the reviewers for their comments and suggestions.

As this came up a few times, we would like to stress to the reviewers that we do not claim that QUIC-FL is generally more accurate than EDEN in practice, but rather QUIC-FL offers two improvements of vital importance:

(1) It has an asymptotically faster decoding time.

(2) It has comparable accuracy in practice (especially for $b>1$), and it has stronger worst-case accuracy guarantees when using the computationally efficient RHT.

We also acknowledge that QUIC’s BSQ (which includes sending a fraction of the coordinates accurately) requires additional bandwidth, and we do factor this into our experiments for a fair comparison.

We further respond to individual comments below.

---

### Meta-Review · Area_Chair_Gpq6 · 2023-12-09

**Metareview:**

This is a truly borderline paper, that makes a contribution in designing unbiased compressor for federated learning (it has little to do with actual *learning* rather distributed mean estimation with limited communication perhaps is more appropriate goal, however this applies to many compression papers in FL). In light of many related compression papers in the past few years, it seems like the impact of this paper will be minor. The compression gain over existing methods seem to be minor, whereas the complexity is suboptimal to QSGD which is a straightforward method for compression (with practically no decoding). This is reflected in the reviews (we have neglected some incorrect comments in a review as the authors point out). After careful consideration, this does not meet the bar of a top conference.

**Justification For Why Not Higher Score:**

Low impact.

**Justification For Why Not Lower Score:**

NA

---

### Decision · Program_Chairs · 2024-01-16

Reject